DOI: 10.1038/s41467-018-06222-0　　**OPEN**

# Single cell molecular alterations reveal target cells and pathways of concussive brain injury

Douglas Arneson[1,2], Guanglin Zhang[1], Zhe Ying[1], Yumei Zhuang[1], Hyae Ran Byun[1], In Sook Ahn[1], Fernando Gomez-Pinilla[1,3,4] & Xia Yang [1,2,5,6]

The complex neuropathology of traumatic brain injury (TBI) is difficult to dissect, given the convoluted cytoarchitecture of affected brain regions such as the hippocampus. Hippocampal dysfunction during TBI results in cognitive decline that may escalate to other neurological disorders, the molecular basis of which is hidden in the genomic programs of individual cells. Using the unbiased single cell sequencing method Drop-seq, we report that concussive TBI affects previously undefined cell populations, in addition to classical hippocampal cell types. TBI also impacts cell type-specific genes and pathways and alters gene co-expression across cell types, suggesting hidden pathogenic mechanisms and therapeutic target pathways. Modulating the thyroid hormone pathway as informed by the T4 transporter transthyretin *Ttr* mitigates TBI-associated genomic and behavioral abnormalities. Thus, single cell genomics provides unique information about how TBI impacts diverse hippocampal cell types, adding new insights into the pathogenic pathways amenable to therapeutics in TBI and related disorders.

[1] Department of Integrative Biology and Physiology, University of California, Los Angeles, Los Angeles, CA 90095, USA. [2] Bioinformatics Interdepartmental Program, University of California, Los Angeles, Los Angeles, CA 90095, USA. [3] Department of Neurosurgery, University of California, Los Angeles, Los Angeles, CA 90095, USA. [4] Brain Injury Research Center, University of California, Los Angeles, Los Angeles, CA 90095, USA. [5] Institute for Quantitative and Computational Biosciences, University of California, Los Angeles, Los Angeles, CA 90095, USA. [6] Molecular Biology Institute, University of California, Los Angeles, Los Angeles, CA 90095, USA. Correspondence and requests for materials should be addressed to F.G-P. (email: fgomezpi@ucla.edu) or to X.Y. (email: xyang123@ucla.edu)

Traumatic brain injury (TBI) is common in domestic, sports, and military environments and often leads to long-term neurological and psychiatric disorders[1]. The hippocampus is a member of the limbic system and plays a major role in learning and memory storage. As a major aspect of the TBI pathology[2], hippocampal dysfunction leads to memory loss and cognitive impairment. The hippocampal formation encompasses four Cornu Ammonis (CA) subfields largely composed of pyramidal cells, and their connections with dentate gyrus (DG) cells. The CA—DG circuitry has served as a model to study synaptic plasticity underlying learning and memory. Glial cells are vital to the hippocampal cytoarchitecture, however, their interactions with neuronal cells are poorly defined. The heterogeneous properties of the hippocampal cytoarchitecture have limited the understanding of the mechanisms involved in the TBI pathology.

Mild TBI (mTBI) is particularly difficult to diagnose given its broad pathology, such that there are no accepted biomarkers for mTBI[3]. This limitation becomes an even more pressing issue given the accumulating clinical evidence that mTBI poses a significant risk for neurological and psychiatric disorders associated with the hippocampus such as Alzheimer's disease (AD), chronic traumatic encephalopathy (CTE), post-traumatic stress disorder (PTSD), epilepsy, and dementia[4]. Accordingly, there is an urgent need to identify functional landmarks with predictive power within the hippocampus to address current demands in clinical neuroscience.

Given that gene regulatory programs determine cellular functions, scrutiny of large-scale genomic changes can reveal clues to the molecular determinants of mTBI pathogenesis including cellular dysfunction, injury recovery, treatment response, and disease predisposition. However, existing genomic profiling studies of mTBI are based on heterogeneous mixtures of cell conglomerates[5–9] which mask crucial signals from the most vulnerable cell types. Here, we report the results of a high throughput parallel single cell sequencing study, using Drop-seq, to capture mTBI-induced alterations in gene regulation in thousands of individual hippocampal cells in an unbiased manner. We focus on concussive injury, the most common form of mTBI, using a mild fluid percussion injury (FPI) mouse model which induces identifiable hippocampal-dependent behavioral deficits despite minimal cell death[10]. We examine the hippocampus at 24 h post-mTBI, as this is a pivotal timeframe for pathogenesis and is generally used for diagnostic and prognostic biomarker discovery[11].

To our knowledge, this is the first single cell sequencing study to investigate the mTBI pathogenesis in thousands of individual brain cells in parallel, offering a cell atlas of the hippocampus under both physiological and pathological conditions. In doing so, we provide novel evidence about the cellular and molecular remodeling in the hippocampus at the acute phase of TBI and help answer critical longstanding questions. Which cell types are vulnerable to mTBI at the acute phase? Within each cell type, which genes have altered transcriptional activities that are induced by mTBI? Which molecular pathways are perturbed by mTBI in each cell type and how do they relate to mTBI pathology and pathogenesis of secondary brain disorders such as AD and PTSD? How do the coexpression patterns of genes across cells and circuits vary in response to mTBI? Through answering these questions, the identified sensitive cell types and associated gene markers can serve as signatures of mTBI pathology that inform on the stage, functional alterations, and potential clinical outcomes. Since the cell is the elementary unit of biological structure and function, we reveal fundamental information that can lead to a better understanding of the mechanistic driving forces for mTBI pathogenesis and identify potential pathways for therapeutic targeting in an unbiased manner. As a proof of concept, we use the data-driven single cell information to prioritize *Ttr*, encoding transthyretin, and show for the first time that modulating *Ttr* improves behavioral phenotypes and reverses the molecular changes observed in mTBI.

## Results

**Unbiased identification of cell identities in hippocampus.** Using Drop-seq[12], we sequenced 2818 and 3414 hippocampal cells from mTBI and Sham animals, respectively. A single-cell digital gene expression matrix was generated using Drop-seq Tools[12] and subsequently projected onto two dimensions using t-distributed stochastic neighbor embedding (t-SNE)[13] to define cell clusters (Methods). We detected 15 clusters each containing cells sharing similar gene expression patterns (Fig. 1a). The cell clusters were not due to technical or batch effects (Supplementary Fig. 1).

To resolve the cell-type identities, we obtained cluster-specific gene signatures (Supplementary Data 1) and compared them to known signatures of hippocampal cell types derived from Fluidigm-based single cell studies[14,15] (Methods). We recovered all known major cell types including neurons, oligodendrocytes, oligodendrocyte progenitor cells (oligoPCs), microglia, mural cells, endothelial cells, astrocytes, and ependymal cells (Fig. 1b). Previously known cell markers, such as *Aqp4* for astrocytes, *Mog* for oligodendrocytes, and *C1qc* for microglia, all showed distinct cluster-specific expression patterns, confirming the reliability of our data-driven approach in distinguishing cell types (Fig. 1c–e; additional examples in Supplementary Fig. 2). Beyond retrieving known cell markers, we also identified novel marker genes for each hippocampal cell type, such as *Calml4* for ependymal, *Vcan* for oligodendrocytes, and *Ly6a* for endothelial cells (Fig. 1f; Supplementary Data 1). We confirmed the cellular identity and hippocampal localization of many novel marker genes using the in situ hybridization (ISH) data from the Allen Brain Atlas (Fig. 2; Supplementary Figs. 3–9).

We identified two potential novel cell clusters whose gene expression signatures did not significantly overlap with any of the known hippocampal cell types, and were hence named Unknown1 (113 cells; 1.8% of all cells tested) and Unknown2 (74 cells; 0.9%; Fig. 1a). Unknown1 has markers indicative of cell growth and migration such as *Ndnf, Nhlh2, Reln*, and *Igfbpl1*, as well as endothelial markers (Fig. 1b), suggesting the cells are migrating endothelial cells. Endothelial cells as main components of the blood brain barrier which is disrupted after mTBI[16], with proliferation and migration of endothelial cells being an intrinsic aspect of new vessel formation[17]. Unknown2 expresses unique markers indicative of cell differentiation such as *Pcolce, Col1a2, Asgr1, Serping1*, and *Igf2*, along with markers of endothelial, mural, and ependymal cells (Fig. 1b), suggesting that these may be progenitor cells differentiating into multiple lineages. To further resolve the identities of these unknown clusters, we examined the expression patterns of their signature genes in the Allen Brain Atlas ISH data. The marker genes of the Unknown2 cluster colocalized to a population of cells in the choroid plexus distinct from the ependymal cells (Fig. 2; Supplementary Fig. 9). The localization of these cells to an area implicated to house stem cells and progenitor cells[18] in conjunction with the functions of the marker genes, supports that this cluster may represent progenitor cells. These cell clusters may illustrate cell types that have been previously left undetected by classical morphological categorization, although detailed functional characterization is required to confirm their identities and functions.

To further characterize neuronal diversity within the hippocampus, we took the cell cluster that expresses canonical neuronal markers (Fig. 1a) and further refined nine subclusters using the

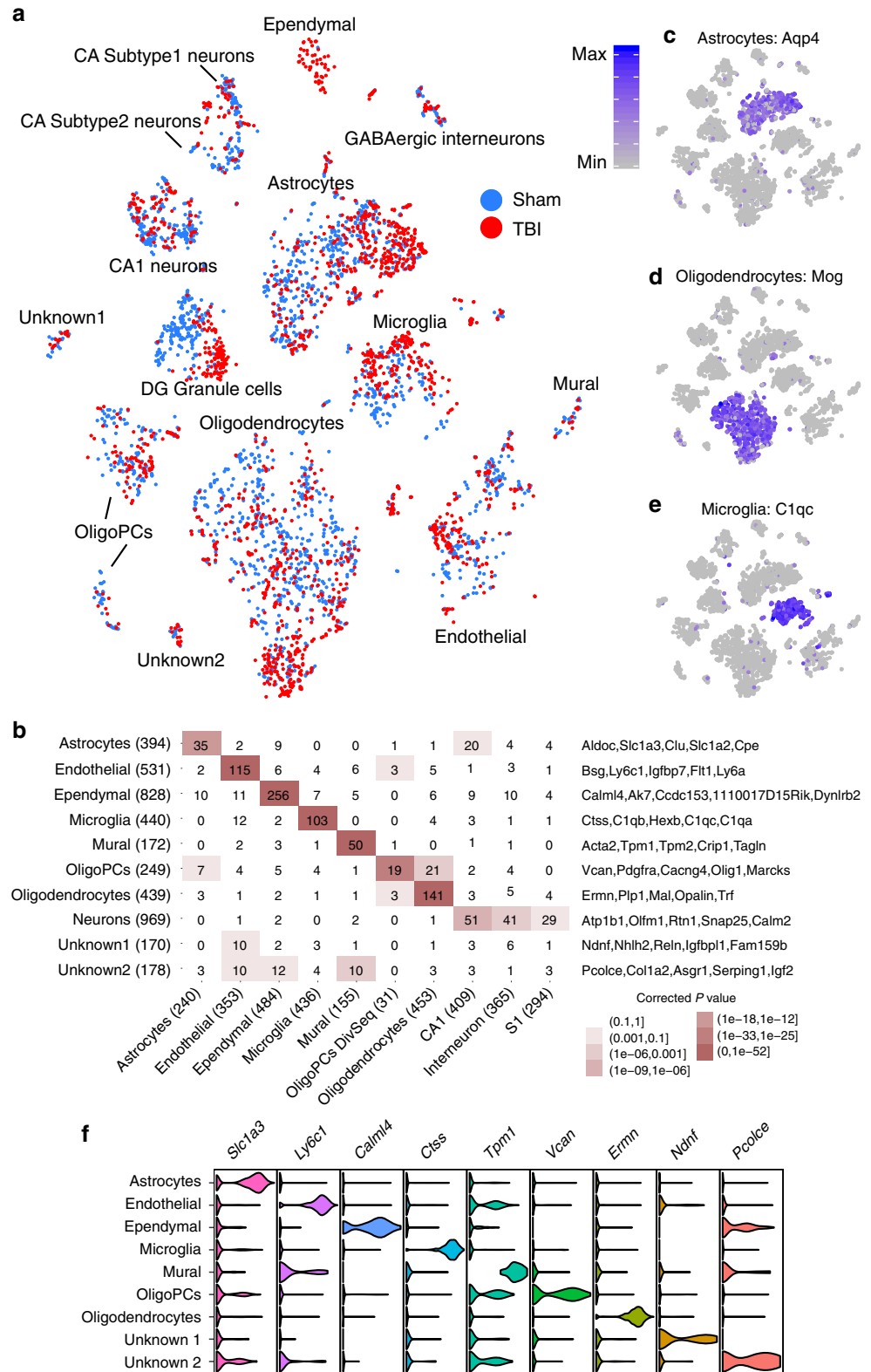

BackSPIN biclustering method which offers better resolution[14] (Fig. 3a). Annotation with known neuronal markers helped resolve GABAergic interneurons, DG granule cells, and 4 subtypes of CA pyramidal neurons (Fig. 3b). However, two clusters (Neuronal Subtype1, Neuronal Subtype 2) express general neuronal markers, but do not express markers of any currently established neuronal subtype in the hippocampus. The functions of their markers (Fig. 3b) suggest that these clusters may contain neurons with the potential to differentiate or self-renew. The unique ability of Drop-seq to catalog cells based on unbiased genomic parameters was also instrumental in unveiling novel neuronal markers[14,15] including *Ptn* in CA1 neurons, *Ly6e* in

**Fig. 1** Determination of major hippocampal cell types and cell type-specific gene markers. **a** t-SNE plot showing cell clusters. Each colored dot is a cell, with blue cells originating from Sham animals and red cells originating from mTBI animals. **b** Overlap between Drop-seq defined marker genes of major cell clusters (rows) with known cell type markers (columns) derived from a previous Fluidigm-based single cell study[14]. Signature marker numbers are indicated in the parenthesis. Fisher's exact test is used to test enrichment with Bonferroni adjusted $p$ values reported. Statistical significance of overlap is indicated by color (the darker the more significant) and the numbers of overlapping genes between our Drop-seq defined markers, and previously known markers are shown in the cells. Top cell marker genes determined by our Drop-seq data are listed on the right of the plot. **c–e** Cluster-specific expression of known cell markers: Astrocytes—*Aqp4*, Oligodendrocytes—*Mog*, and Microglia—*C1qc*. This analysis confirms that each cluster captures a particular cell type. **f** Normalized expression values of top cell type-specific marker genes are plotted as violin plots with cell types as rows and genes as columns. Cells were from 3 sham and 3 mTBI animals

CA3 neurons, and *Sema5a* in DG granule cells (Fig. 3b; full list in Supplementary Data 1). We confirmed the subtype-specific expression patterns in hippocampal subregions and cell types (Fig. 3c–e) and further verified the expression specificity of several novel markers using the Allen Brain Atlas ISH data (Fig. 2, Supplementary Figs. 3–9). The CA Subtype2 cells were found to be located in the Subicular Complex (Fig. 2), which mediates the main output of signals from the hippocampus. Like the CA subregions, the Subicular Complex is also comprised of pyramidal neurons, which explains the expression of CA neuronal signature genes in addition to the genes specific to this cell cluster. The subiculum is involved in pathology of CTE and the associated dementia following TBI[4].

These results indicate that our transcriptome-driven, unbiased Drop-seq approach has the unique ability to uncover potential new cell types, states, and markers based on genomic features that may infer function. Based on the central dogma, gene regulation is upstream of the production of proteins, which are fundamental for cell function. In contrast, morphology-based approaches may not offer the resolution to distinguish subtypes of cell populations that share similar morphology but carry certain unique functions. For instance, cells of the two CA subtype clusters uniquely express specific marker genes which may infer unique functions. However, detailed functional characterization of these potential new cell subtypes is required in future studies to test functional differences and their role in the hippocampal circuitry under homeostatic and/or pathological conditions.

**Identification of vulnerable hippocampal cell types to mTBI.** To retrieve hippocampal cell types vulnerable to mTBI, we first compared the cell population proportions between mTBI and Sham animals. Ependymal cells were found to be more abundant in mTBI compared to Sham animals (89% from mTBI samples vs 11% from Sham samples, Fig. 1a). This large shift in relative abundance of ependymal cells at 24 h post-surgery is consistent with the reported role of ependymal cells in acute post-injury processes such as circuit repair, scar formation, and potential source of progenitor cells[19,20]. We also see significant increases in the cell proportions of microglia, endothelial cells, and Unknown 2, as well as significant decreases in neurons, oligodendrocytes and oligoPCs post-TBI (Supplementary Table 1). However, many experimental factors in the Drop-seq procedure may influence the capture rate of different cell types between samples, and changes in the relative proportion of a cell type do not directly implicate cell proliferation or death but could be the result of shifts in other cell types. Therefore, caution is needed in the interpretation of these results.

We also examined the shifts in global transcriptome patterns within each cell type by testing if the Euclidean distance of gene expression profiles of cells between the two groups is larger than expected by chance (details in Methods). This analysis revealed that DG, ependymal cells, astrocytes, microglia, oligodendrocytes, oligoPCs, endothelial cells, Neuronal Subtype 1, CA1, CA

Subtype1, and GABAergic neurons demonstrate significant global transcriptomic shifts between mTBI and Sham (Supplementary Fig. 10). In particular, mTBI had such a profound effect on the transcriptome of DG granule cells, that they became clearly separated into two distinct clusters (Figs. 1a, 3a). Post-traumatic epilepsy is a major concern in TBI and is attributable to DG dysfunction[21].

The above analyses implicated the majority of hippocampal cell types to be vulnerable to mTBI and open the possibility for investigating the specific roles of each cell type in mTBI pathology and for designing targeted treatments. For example, the genomic markers of the DG granule cells highly sensitive to mTBI can be used to design treatments that target specific DG subpopulations responsible for post-traumatic epilepsy to avoid the side effects of classical epilepsy treatments targeting broad populations of cells.

**mTBI alters cell-cell gene co-expression in the hippocampus.** Emerging evidence in the neuroimaging field suggests that changes in the interaction patterns among cells in circuits can contribute to reduced cognitive capacity in TBI[22]. We used the co-regulation patterns between genes of different cell types to infer interactions among cells, as gene co-expression can infer functional connectivity[23,24]. Specifically, we focused on marker genes encoding secreted peptides from each cell type (source cells), which have the potential to interact with receptors and regulate downstream effector genes in other cell types (target cells) (Methods; Fig. 4a). This approach was found to recapitulate known cell–cell communications within the trisynaptic circuit of the hippocampus (Fig. 4b–d). The gene co-expression patterns reflected the known mutual interactions between DG and CA3 as well as the CA3 to CA1 interaction. We applied our method to the mTBI data and found extensive reorganization in the pattern of gene coexpression, potentially representing interactions, among cells in response to mTBI. For example, the interaction from astrocytes and ependymal cells to neurons and from microglia to oligodendrocytes was enhanced in mTBI (Fig. 4e, f). We also found decreased interaction between microglia and neurons, and decreased interaction from oligodendrocytes to neurons in mTBI. These shifts in the pattern of gene co-regulation among hippocampal cell types may implicate reorganization of the working flow in response to mTBI challenge. The reorganization was also reflected in changes in the association pattern among single genes. For instance, the correlation of *Bdnf* from neurons with cell metabolism genes in microglia was lost in mTBI. Notably, the known AD risk gene *Apoe* in astrocytes and ependymal cells (source cells) showed strong correlations with mitochondrial metabolism genes in neurons (target cells) after mTBI. These results suggest the potential of mTBI to alter the interactive patterns at the level of circuits and genes in the hippocampus, with cell metabolism involved in the interactions (detailed peptide-gene correlations in Supplementary Data 2). We acknowledge that this in silico analysis points towards potential circuit perturbations but would require downstream validation.

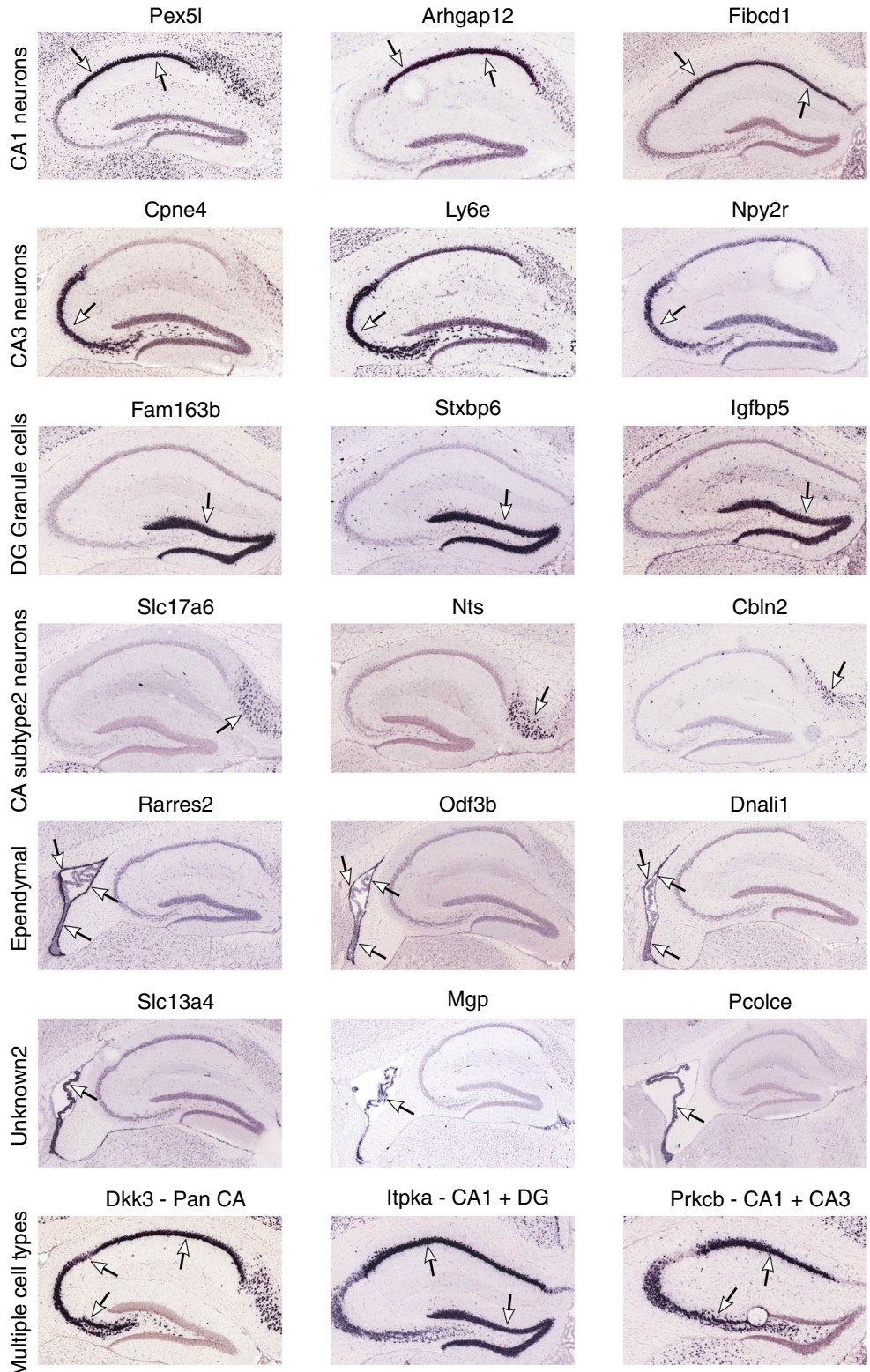

**Fig. 2** Cross-validation of novel marker genes for specific neuronal subpopulations. To validate the specificity of novel marker genes for neuronal populations and to help resolve the identity of previously unknown cell clusters, we examined the expression patterns of our cell markers in the ISH images from the Allen Brain Atlas[58]. Here, we showcase three select novel genes from four cell types: CA1 neurons, CA3 neurons, DG granule cells, and ependymal cells. Additionally, we showcase marker genes expressed across multiple cell types, genes which resolve the Unknown2 cluster to cells inside the choroid plexus, and genes which help resolve CA Subtype2 Neurons to the Subicular Complex

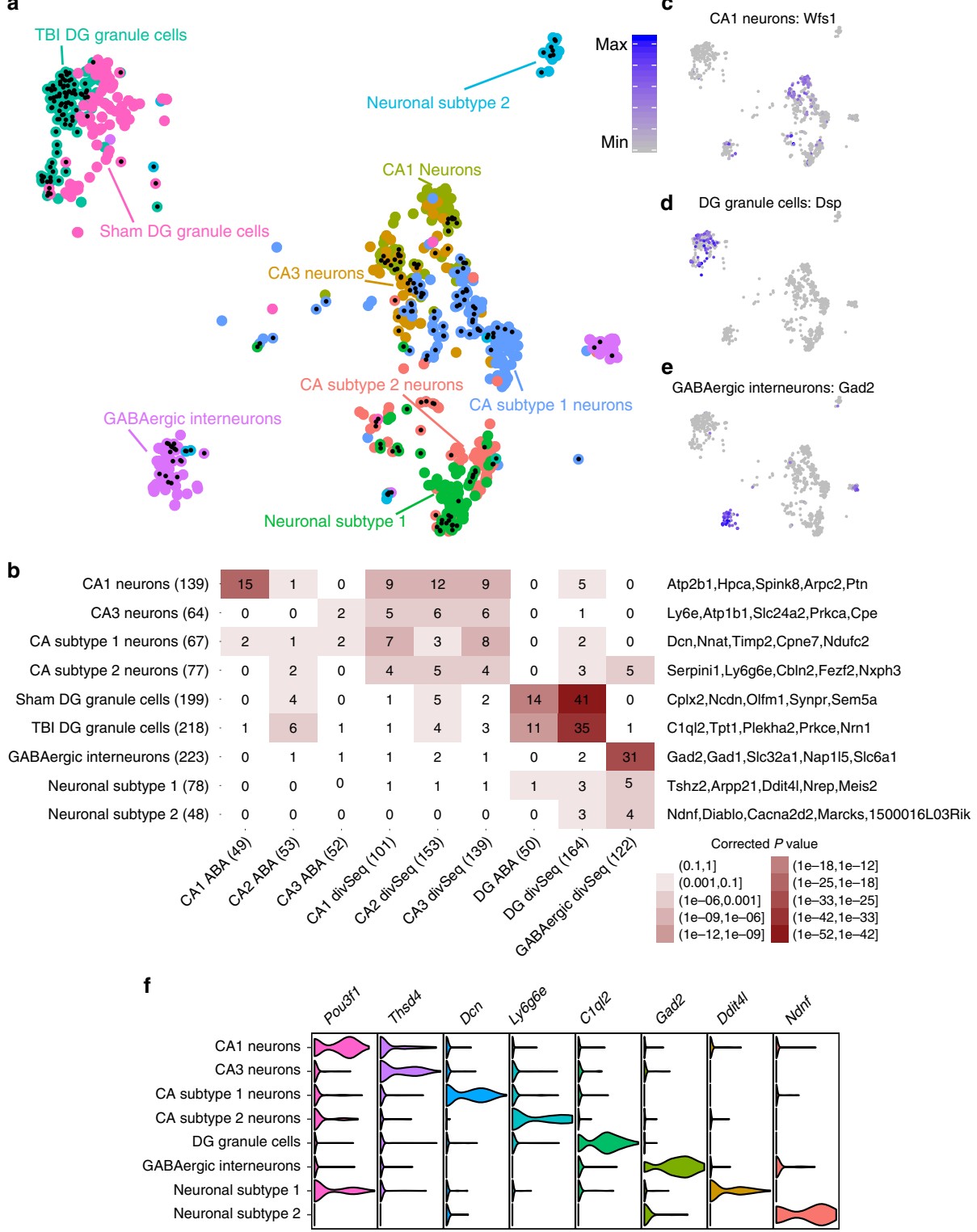

**Fig. 3** Determination of neuronal cell subtypes and cell type-specific gene markers. **a** t-SNE plot of neuronal subtypes determined by backspin biclustering. Each color indicates a different cell type cluster identified, and cells with a black dot at their center are from TBI samples. **b** Overlap of Drop-seq defined marker genes of the neuronal subtypes (rows) with those of the previously defined hippocampal neuronal cell types (columns). Known markers were derived from Alan Brain Atlas (ABA)[58] and Habib et al. using Div-Seq[15]. Signature marker numbers are indicated in the parenthesis. Fisher's exact test is used to test enrichment with Bonferroni adjusted $p$ values reported. Statistical significance of overlap is indicated by color (the darker the more significant), and the numbers of overlapping genes between our Drop-seq defined markers and previously known markers are shown in the cells. Top cell marker genes determined by our Drop-seq data are listed on the right of the plot. **c–e** Cluster-specific expression of known cell markers: CA1 neurons—*Wfs1*, DG granule cells—*Dsp*, and GABAergic interneurons—*Gad2*. **f** Normalized expression values of top neuronal subtype-specific marker genes are plotted as violin plots with cell types as rows and genes as columns

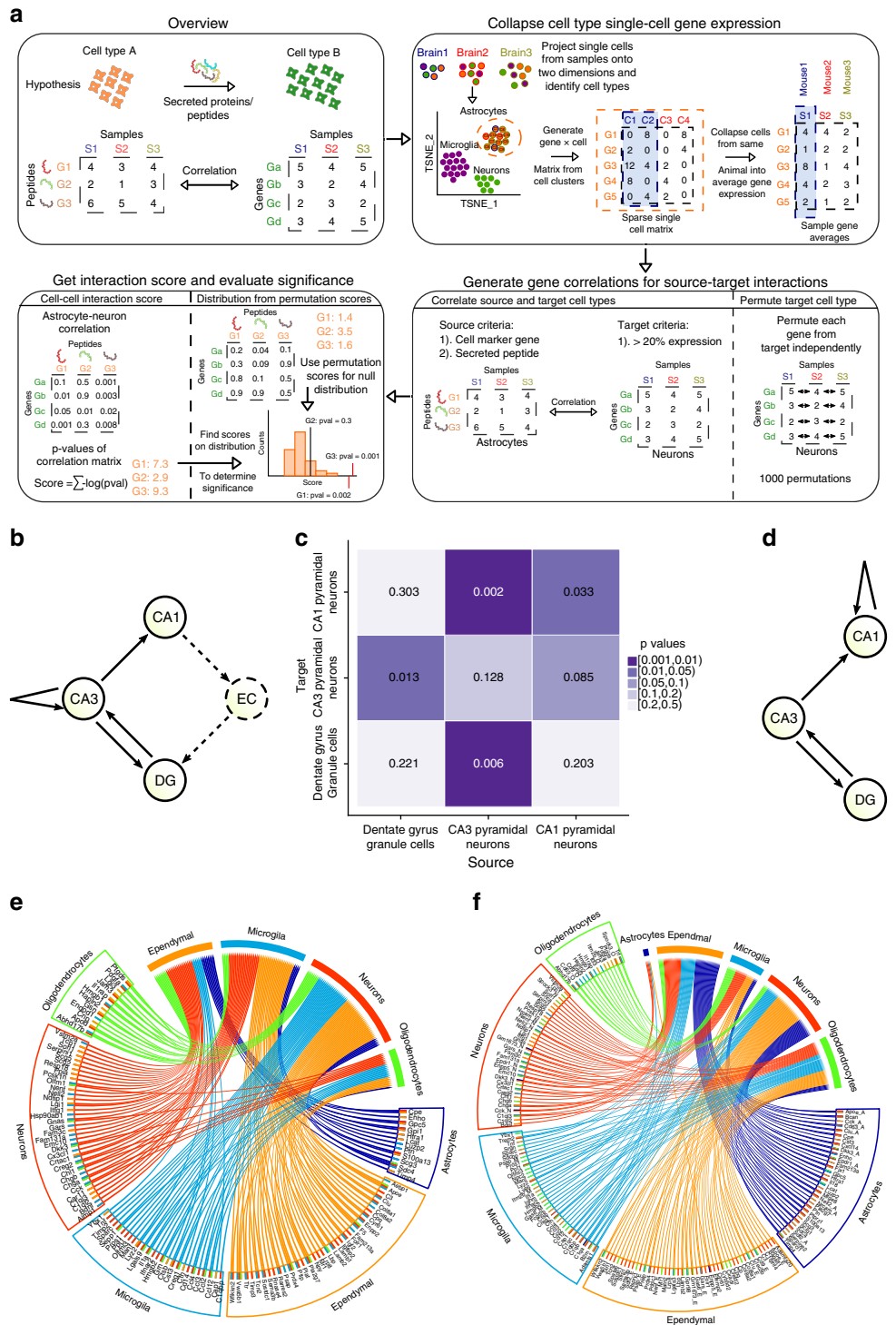

**Identification of genes and pathways vulnerable to mTBI.** To determine specific genes and pathways that may confer mTBI pathogenesis in each hippocampal cell type and to further refine the vulnerable cell types, we identified differentially expressed genes (DEGs) between Sham and mTBI within each cell cluster (Table 1; all DEGs in Supplementary Data 3) at false discovery rate (FDR) <0.05. Astrocytes, oligodendrocytes, and neurons had the largest numbers of DEGs between mTBI and Sham samples (Table 1). Annotation of the DEGs with curated biological pathways from KEGG[25], Reactome[26], BIOCARTA, and Gene Ontology[27] databases identified key pathways that could explain

fundamental aspects of the mTBI pathology and the main cell types involved (Table 1; full pathway list in Supplementary Data 4). For example, our results showed diverse pathways to be involved in mTBI, encompassing energy and metabolism (astrocytes, neurons), inflammation and immune response (microglia, oligodendrocyte PCs), myelination (oligodendrocytes), amyloids (endothelial and ependymal cells), neurogenesis and synaptic signaling (neurons), cell migration (GABAergic interneurons), glutamate transport (CA1 pyramidal cells), regulation of cell death (astrocytes and microglia, multiple neuronal populations), and dendrite morphogenesis (Unknown 1 cluster). Our data

**Fig. 4** TBI alters cell-cell gene co-expression in the hippocampus. **a** Cell-cell gene co-expression analysis method. Secreted proteins or peptides from a source cell can communicate with genes in a target cell, which can be captured by strong correlations between the secreted proteins in the source cells and genes in the target cells. For each cell cluster, the expression of each gene was summarized to individual animal level, and a correlation matrix between genes of different cell types was constructed. The -log10 p-values of the correlations for each secreted peptide are summed to obtain an interaction score of that peptide with a particular target cell type. The cell type gene expression matrix is then permuted to generate the null distribution of interaction scores to calculate the significance of observed interaction scores. **b** Schematic of known cell type interactions within the trisynaptic circuit. The entorhinal cortex (EC) is not captured in our single cell analysis so we cannot validate its edges (dashed). **c** Heatmap displaying the permutation-based p-values of glutamate-driven cell–cell gene coexpression; columns are source cells and rows are target cells. Darker purple indicates significant (lower p value) interactions. **d** Redrawn cell-cell interaction schematic based on our cell–cell gene coexpression method. All between-cell type interactions among CA1, CA3, and DG cells are recapitulated, but within cell type self-interactions differ from the known schematic in **b**. **e, f** Circos plots of significant cell-cell gene coexpression among hippocampal cell types in Sham (**e**) and TBI groups (**f**). The bottom half of each circos plot shows source cell types with secreted peptides listed and the top half are the target cell types (genes not shown because many genes show strong coexpression; listed in Supplementary Data 2). Colored lines in the center indicate significant connections of the peptides with different cell types. Comparison between **e** and **f** shows reorganization in the gene coexpression patterns among hippocampal cell types after TBI and the genes potentially driving the interactions. As cell transcriptome can instruct cell-cell communication to process high order information, these results suggest potential changes in neural circuit organization after an episode of TBI

uniquely points to the specific cell types engaging these pathways and offers novel insights into the functions of individual cell types, including previously understudied cell populations, in mTBI pathogenesis.

TBI is followed by a stage of metabolic dysfunction[28] which reduces the capacity of the brain for activity-dependent plasticity[29]. Our results provide a better understanding of the molecular and cellular underpinnings of the metabolic crisis typical of mTBI by identifying down-regulation of mitochondrial metabolic genes in astrocytes and CA1 pyramidal cells. Astrocytes supply energy substrates to neurons and are essential for neuronal function. Interestingly, our results indicate that CA1 pyramidal cells also experience metabolic alterations at this early stage of mTBI pathogenesis (24 h). In addition, DEGs in CA1 pyramidal cells informed on increased expression of glutamate transporters, which could explain the altered capacity to sustain long-term potentiation post-TBI[30].

The risks posed by mTBI on the development of other neurological disorders are a pressing issue in clinical neuroscience. The genes, pathways, and the associated cell types that we identified provide insights into the molecular and cellular bases for these disease connections. DG granule cells, which function to interact with CA pyramidal cells (Fig. 4b), showed alterations in genes involved in cell-cell signaling (*Npy, Penk, Ptprn,* and *Ihnba*) as well as neuroplasticity genes such as *Bdnf* and *Ntf3*. These pathways may underlie the sensitivity of DG granule cells to contribute to post-traumatic epilepsy after mTBI. Pathways informed by DEGs from the endothelial and ependymal cells implicate the importance of these cell types in amyloid buildup, as indicated by the upregulation of the known pro-amyloid deposition gene *B2m* and down-regulation of inhibitors of beta-amyloid aggregation (*Apoe, Itm2a, Itm2b,* and *Itm2c*) in these cell types. As dysregulated metabolism and amyloid deposition are key features in AD, CTE, and PD[31], our study provides detailed information on the specific cell types such as astrocytes, CA1 pyramidal cells, endothelial and ependymal cells that could be the starting loci for the wave of post-mTBI amyloid buildup in chronic mTBI.

**mTBI alters gene expression in a cell-type specific manner.** We found that many of the DEGs were significantly altered by mTBI in a specific cell type (Fig. 5a, b) and showed clear cell type-specific shifts in expression patterns (Fig. 5c–e). Importantly, >50% of the DEGs identified at the single cell level were masked in bulk tissue analysis (Fig. 5f; tissue-level DEGs in Supplementary Data 3) and these unique cell-level DEGs were primarily

from cell types of low abundance such as neuronal subpopulations. On the other hand, the common DEGs between single-cell and tissue-level analyses were mainly from abundant cell types such as astrocytes and oligodendrocytes. As less abundant cells such as neurons carry essential functions in the brain, the cell type-specific DEGs can be strong drivers of disease but will be missed in bulk tissue analysis. Therefore, genomic information in individual cell types has the advantage to extract hidden mechanisms involved in TBI pathology that otherwise would be masked in bulk tissue studies.

Cell type-specific DEGs may serve as selective biomarkers or therapeutic targets that can trace or normalize specific abnormalities of mTBI pathology (Fig. 6a). For instance, *Id2*, a gene previously described to be upregulated by seizures in DG[32], is specifically upregulated by mTBI in DG granule cells and could serve as a potential target to temper post-traumatic epilepsy. We also found that *P2ry12*, a marker for brain resident microglia[33], is specifically downregulated in microglia post-mTBI, suggesting the potential utility of *P2ry12* as a marker for early inflammatory response to TBI. Interestingly, *Trf*, encoding transferrin for iron transport was upregulated in oligodendrocytes, suggesting a possible involvement of *Trf* on the described association between iron deposition and cognitive deficits in mTBI[34,35]. Several of the cell-type specific DEGs are related to amyloid deposition and AD, including *Apoe*—an ependymal-specific DEG and known for its effects on AD and TBI[36], and *Itm2a*—an endothelial specific DEG and an inhibitor of amyloid-beta production and deposition[37]. These results indicate that the putative action of TBI in amyloid regulation involves various cell types.

We also found several cell-type specific mTBI responsive genes that have not been implicated in mTBI previously. For instance, a CA1 pyramidal cell-specific DEG *Klhl2*, which encodes an actin binding protein, was recently implicated in human neuroticism[38,39] and showed increased neuronal expression post-mTBI (Fig. 6a). *Klhl2* in CA1 pyramidal cells may serve as a novel link between mTBI and the increased tendency for neuroticism observed in blast TBI[40]. *Arhgap32* is primarily upregulated in CA subtype 2 cells in mTBI. It encodes a neuron-associated GTPase-activating protein that may regulate dendritic spine morphology and strength[41]. *Arhgap32* may be critical for supporting transmission of information across hippocampal cells. The endothelial-specific gene *Fxyd5* encodes a glycoprotein that enhances chemokine production and inhibits cell adhesion by downregulating E-cadherin[42]. It was upregulated in mTBI in endothelial cells, suggesting that it may play a role in neuroinflammation and blood-brain-barrier dysfunction associated with TBI.

**Table 1 Top enriched pathways among DEGs of major cell types (FDR<5%) and representative DEGs in the select pathways**

| Cell type | No. DEGs | Top DEG pathways | Top 5 representative DEGs |
|---|---|---|---|
| Astrocytes | 247 | Metabolic depression | Down: Mdh1,Atp5b,Cox4i1,Atp5a1,Ndufs7 |
|  |  | Calcium/calmodulin pathways | Up: S100a11,Syt1; Down: Calm1,Calm2,Camk4 |
| Microglia | 57 | Inflammation/immune response | Up: Cebpb,Il1b,Cxcl1; Down: Selplg,Cx3cr1 |
| Oligodendrocytes | 115 | Myelination | Up: Klk6; Down: Mbp,Mal,Sirt2,Tspan2 |
|  |  | Oligodendrocyte differentiation | Up: Sox10; Down: Gstp1,Cnp,Tspan2,Olig1 |
| (Oligodendrocyte PCs) | 7 (103) | Myelination | Down: Mbp,Plp1,Sirt2,Mag |
|  |  | Immune response | Up: Prkx; Down: Egr1,Fyn |
| Endothelial | 35 | Amyloids | Up: Ttr,B2m; Down: Itm2a,Itm2b |
| (Ependymal) | 87 (783) | Cilia related pathways | Down: Tmem107,Ift43,Dynll1,Spag17,Spef2 |
|  |  | Platelet degranulation | Up: Igf2; Down: Rarres2,Scg3,Clu,Pros1 |
|  |  | Amyloids | Up: Cst3; Down: Itm2c,Apoe,Bace2,Apbb1 |
| (Unknown1) | 1 (52) | Dendrite morphogenesis | Down: Epha4,Stk11,Baiap2,Bhlhb9 |
| Neurons—general | 139 | Neurogenesis | Up: Npy,Inhba,Adgrl3; Down: Ndn,Cck |
|  |  | Energy-related | Up: Atp1a1,Atp1a2,Atp2b2; Down:Atp1b1 |
|  |  | Synaptic signaling | Up: Scn1b,Cplx2,Slc17a7,Grik4,Grin2b |
| (CA1 neurons) | 16 (330) | Glutamate transport | Up: Slc17a7,Grin2b,Gria1,Gria2 |
|  |  | Metabolic depression | Down: Ndufa4,Atp5d,Atp5g1,Ndufv3,Cox8a |
| (CA3 neurons) | 14 (209) | Biosynthesis | Up: Ncan; Down: Mrpl57,Eef1a2,Farsb,Rpl15 |
| (CA subtype1 neurons) | 6 (204) | Neurotransmitter pathways | Up: Camk2a,Ppp1r1b,Grin2b,Cav2,Prkcg |
| (Neuronal subtype2) | 11 (139) | Proteosome | Up: Rpn1,Psmc6,Psma1 |
| DG granule cells | 44 | Neuroplasticity/neurotropic | Up: Set,Bdnf; Down: Chl1,Ntf3,Pcp4 |
|  |  | Cell-cell signaling | Up: Ptprn,Penk,Npy,Inhba,Pcdh8 |
| (GABAergic interneurons) | 6 (240) | Cell migration | Up: Wasl,Arpc3,Pik3ca |

DEGs listed as up had increased expression in TBI and DEGs listed as down had decreased expression in TBI compared to sham. Cell types shown in parenthesis are rare cell types with fewer cells analyzed, and hence did not have sufficient numbers of DEGs at FDR < 5% between Sham and TBI samples to reveal significant pathways. Instead, DEGs reaching a threshold of $p < 0.01$ (number of DEGs shown in parenthesis) were used to derive suggestive pathways for these rare cell types. P values were determined using a bimodal likelihood ratio test

**Validation of cell-type specific mTBI DEGs.** We used the RNAscope multiplex RNA ISH to validate select cell-type specific mTBI DEGs (selected from Fig. 6a) by co-hybridizing hippocampal tissue slices with a cell identity marker and a DEG for each select cell type. Using $n = 8$ animals per group, over 900 z-stack images with two ISH channels (one for cell markers and the other for DEGs) and a DAPI (for nuclei) channel were generated for multiple DEGs and cell types. RNAscope is a high-resolution quantitative ISH which detects single mRNA molecules. Co-localization between a cell marker gene and a DEG is determined by the presence of fluorescent spots representing single mRNA molecules of both genes within the same cell. Automated cell segmentations were performed using the CellProfiler software[43] and quantification of mRNA molecules per cell was carried out using FISH-quant[44]. To quantify cell type-specific differential expression of a DEG, we first determined the cell type of interest from a z-stack image by thresholding the counts of the cell type marker, followed by comparing the counts of the DEG mRNA molecules only within that cell type between Sham and TBI samples (Methods). As shown in Fig. 7 (high magnification images of select regions) and Supplementary Fig. 11 (low magnification images of larger areas), the select DEGs are localized in the same cell segments expressing the corresponding cell identity markers. Quantitative assessments validated significant differential expression of cell-type specific DEGs in Microglia (P2ry12), DG granule cells (Id2), and CA Subtype 2 neurons (Arhgap32) in both CA1 and CA3 regions ($p < 0.05$ by bimodal likelihood ratio test; $n = 8$/group). Klhl2 in CA1 pyramidal neurons did not reach statistical significance ($p = 0.07$) but showed an increasing trend which is consistent with the Drop-seq data.

**Prioritization of mTBI targets based on single cell data.** The aforementioned cell-type specific genes perturbed by mTBI provide unique information about the microcircuits underlying the mTBI pathophysiology. These can be leveraged for the design of

therapeutic interventions to target specific cell types driving pathological manifestations if their causal roles in pathogenesis are confirmed. Conversely, identifying DEGs that are affected across multiple cell types by mTBI has the potential to pinpoint the most vulnerable genes that are responsible for the broad symptomology of mTBI. Such genes cannot be retrieved without examining multiple cell types individually to confirm the widespread effect across cell types. Given the broad aspects of the TBI pathogenesis, targeting such pan-hippocampal DEGs may offer broader and stronger therapeutic effects by normalizing the functions of multiple cell types.

Based on the consistency of differential expression across cell types, we prioritized a number of multi-cell type DEGs affected by mTBI (Fig. 6b). Several such DEGs are related to beta-amyloid and AD, including Ttr—encoding transthyretin which is an amyloid beta scavenger[45], mt-Rnr2—encoding the mitochondria factor humanin[46] which is protective against AD, Itm2b—an inhibitor of beta-amyloid deposition[47], and Apba2 (Mint2)—a stabilizer of amyloid precursor protein APP[48]. These pan-hippocampal DEGs, along with some of the cell-type specific DEGs previously discussed, strongly implicate pathways related to post-mTBI AD pathogenesis and can be targeted for post-mTBI AD prevention. Slc17a7 is a pan-neuronal DEG. It encodes a sodium-dependent phosphate transporter in neuron-rich regions of the brain and functions in glutamate transport[49]. Interestingly, genetic polymorphisms in this gene have been associated with recovery time and severity of outcomes after sport-related concussion in humans[50].

Notably, the gene Ttr represents the most robust DEG across hippocampal cell types in that it was a top DEG with increased expression post-mTBI in 7 of the 10 major cell types and 6 of the 8 neuronal clusters, with the highest expression and strongest induction seen in ependymal cells (Fig. 6b). Therefore, the full information across hippocampal cell types established Ttr as the most consistent DEG post-mTBI, providing a strong rationale to

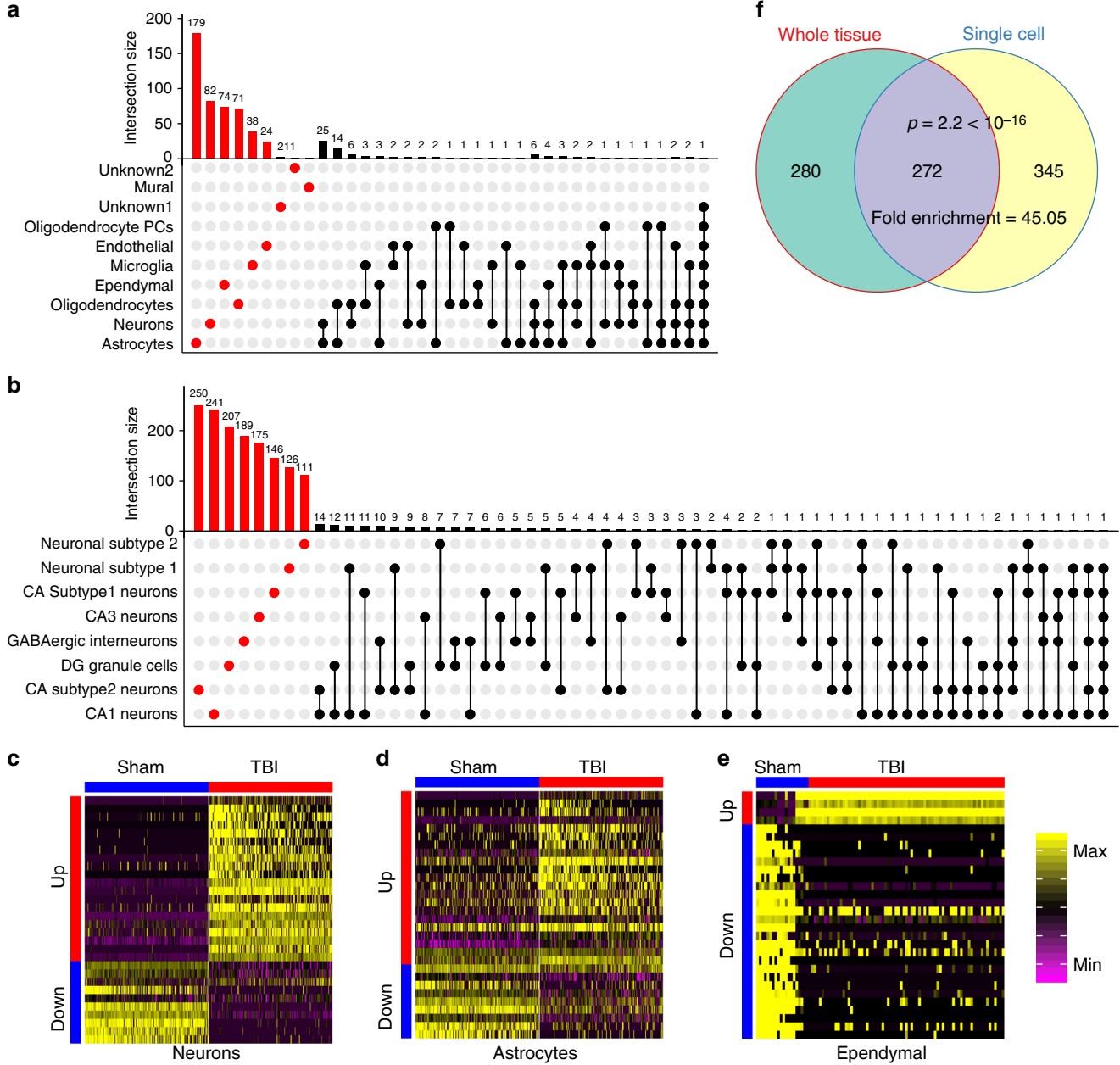

**Fig. 5** Differentially expressed genes (DEGs) induced by TBI in hippocampal cell types. **a**, **b** DEGs unique to a cell type are indicated in red and those shared between ≥2 cell types are indicated by black dots. The histogram above each plot indicates the DEG counts for each category. **a** The majority of the DEGs are cell-type specific. **b** The majority of the DEGs in neuron subtypes are subtype-specific. **c–e** Heatmaps of DEGs in select cell types demonstrate clear differential expression patterns between Sham and TBI cells. **f** Many cell-type specific DEGs cannot be captured in the bulk tissue analysis, supporting the uniqueness of using single cell genomic analysis. DEG overlap p-value was calculated using Fisher's exact test

prioritize *Ttr* for testing. *Ttr* encodes transthyretin, a transporter that carries the thyroid hormone thyroxine, preferentially T4, across the blood brain barrier[51]. We hypothesize that the pan-hippocampal upregulation of *Ttr* might implicate an impaired thyroid hormone pathway in TBI and a compensatory need for thyroxine T4, the major brain-specific substrate of transthyretin. Given the strong implication of altered cell metabolism in various cell types discussed above, the critical role of thyroid hormone in regulating metabolism could serve as a platform to mitigate the metabolic crisis after mTBI across a broad range of hippocampal cell types.

As a proof of concept, we examined the possibility that modulating the thyroid hormone pathway and *Ttr* may serve as an effective strategy to counteract mTBI pathology. We first

quantitatively confirmed, using RNAscope multiplex ISH assays, that *Ttr* was indeed significantly upregulated in multiple cell types and brain regions as predicted by Drop-seq (Fig. 8 for high magnification images, Supplementary Fig. 12 for low magnification images). Acute intraperitoneal injection of T4 within the first 6 h post-mTBI protected learning and memory, as determined by the Barnes Maze test one-week post-mTBI. For the learning component, animals were trained with two trials per day for four consecutive days, and memory retention was assessed two days after the last learning trial. Differences in learning and memory between groups were determined using a two-tailed Student's t-test. The learning effects were evidenced by a sustained reduction in the latency to find the escape hole across all time points in the T4 group compared to TBI mice without T4 (Fig. 9a, b). The

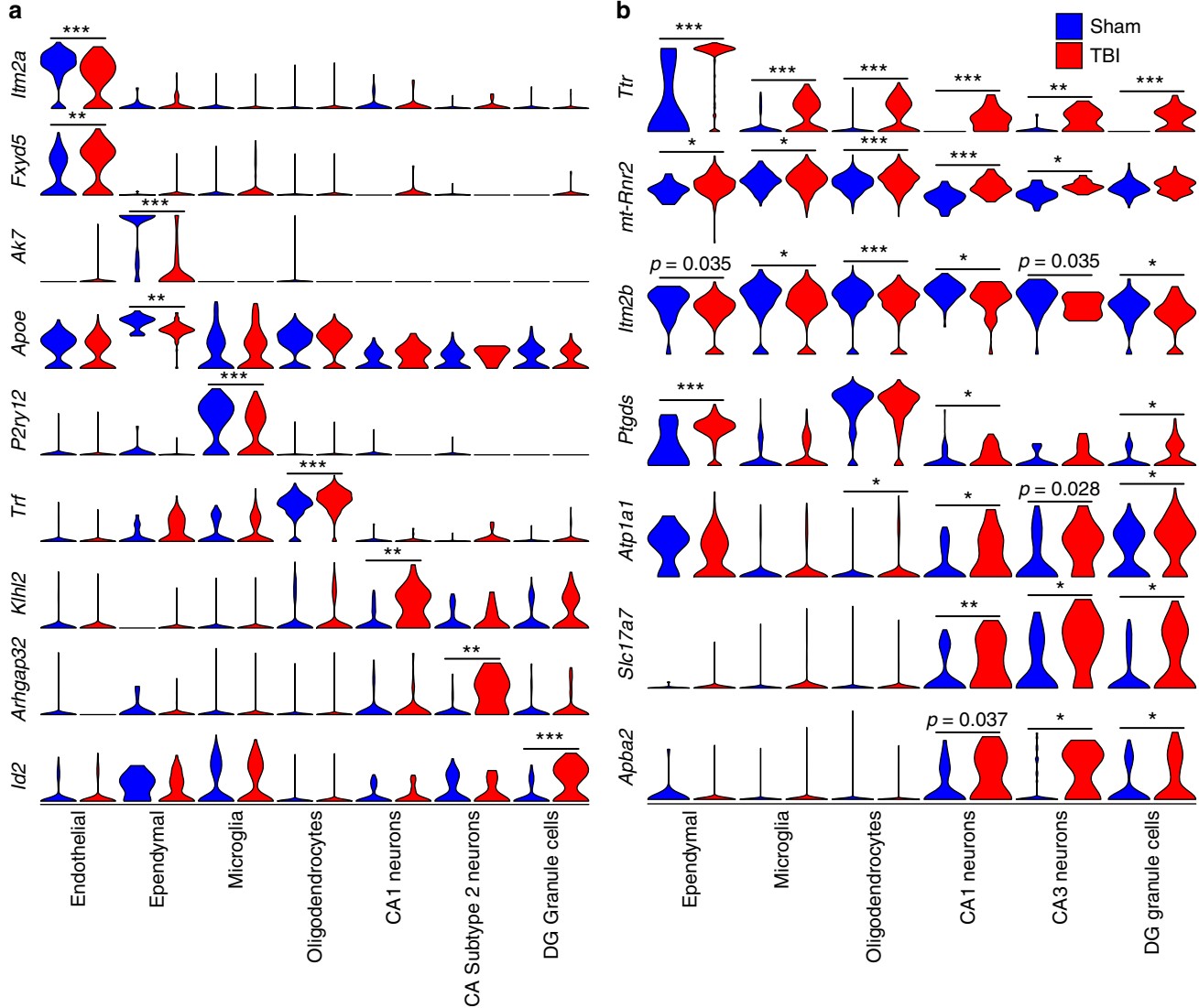

**Fig. 6** Top cell-type specific DEGs and pan-hippocampal DEGs. The normalized expression of cell-type specific (**a**) and pan-hippocampal (**b**) DEGs between Sham and TBI samples is displayed as violin plots. Single cells from Sham samples are indicated by the blue plots and single cells from TBI samples are indicated by the red plots. Likelihood-ratio test was used to determine statistical significance between Sham and TBI groups and FDR was calculated to correct for multiple testing. *FDR < 0.02, **FDR < $1 \times 10^{-4}$, ***FDR < $1 \times 10^{-6}$

effects of T4 on memory were demonstrated by the recovery of the latency time to a level comparable to the Sham group (Fig. 9c, d). The improved Barnes Maze performance was not due to changes in velocity (Supplementary Fig. 13). It is noteworthy that T4 was injected immediately after the lesion, which could protect a variety of cellular functions against the disruptive effects of TBI. This is the first time that T4 treatment was found to mitigate cognitive deficits in a mouse model of concussive injury.

A recent study using cortical impact injury suggested a potentially protective action of T4 by testing candidate genes related to hypoxia and neurogenesis[52]. Our analysis of known T4 transporters indicates that T4 treatment primarily downregulates *Ttr* and has weaker effects on genes encoding other transporters (Fig. 9e). Genes encoding thyroid hormone receptors that are downstream of T4 also had less consistent changes across cell types compared to *Ttr* (Supplementary Table 2). These results agree with our hypothesis that the upregulation of *Ttr* seen in TBI is an indicator of thyroid hormone deficiency in the brain and T4 treatment reverses this change. Although our gene expression

data suggests that *Ttr* is more strongly modulated by T4 compared to the other known T4 transporters and receptors, substrate binding experiments are needed to test whether Ttr is the main T4 transporter. Through whole transcriptome profiling, we also identified a cascade of genes and pathways potentially involved in T4 effects. T4 treatment affected a total of 951 DEGs involved in diverse pathways that could support cell functions ranging from metabolic processes, cell differentiation, to cell–cell signaling (Supplementary Data 5). Among the T4 DEGs, 121 were also altered by mTBI (Fig. 9f) and T4 reversed the TBI-induced changes of 93 genes (Fig. 9g), including *Ttr*, *Wdr72* (implicated in cognitive processing speed[53]), and *Tpx2* (protective of neurocyte apoptosis in an AD model[54]) (Fig. 9h). The 93 mTBI genes normalized by T4 (Fig. 9g) were enriched for hormone response and metabolic pathways (Fig. 9i). However, there are pathways that are affected by TBI but are not reversed by T4 treatment as well as genes affected by T4 treatment but are not perturbed by TBI (Supplementary Table 2; Fig. 9i), suggesting that T4 normalizes select aspects of TBI pathology.

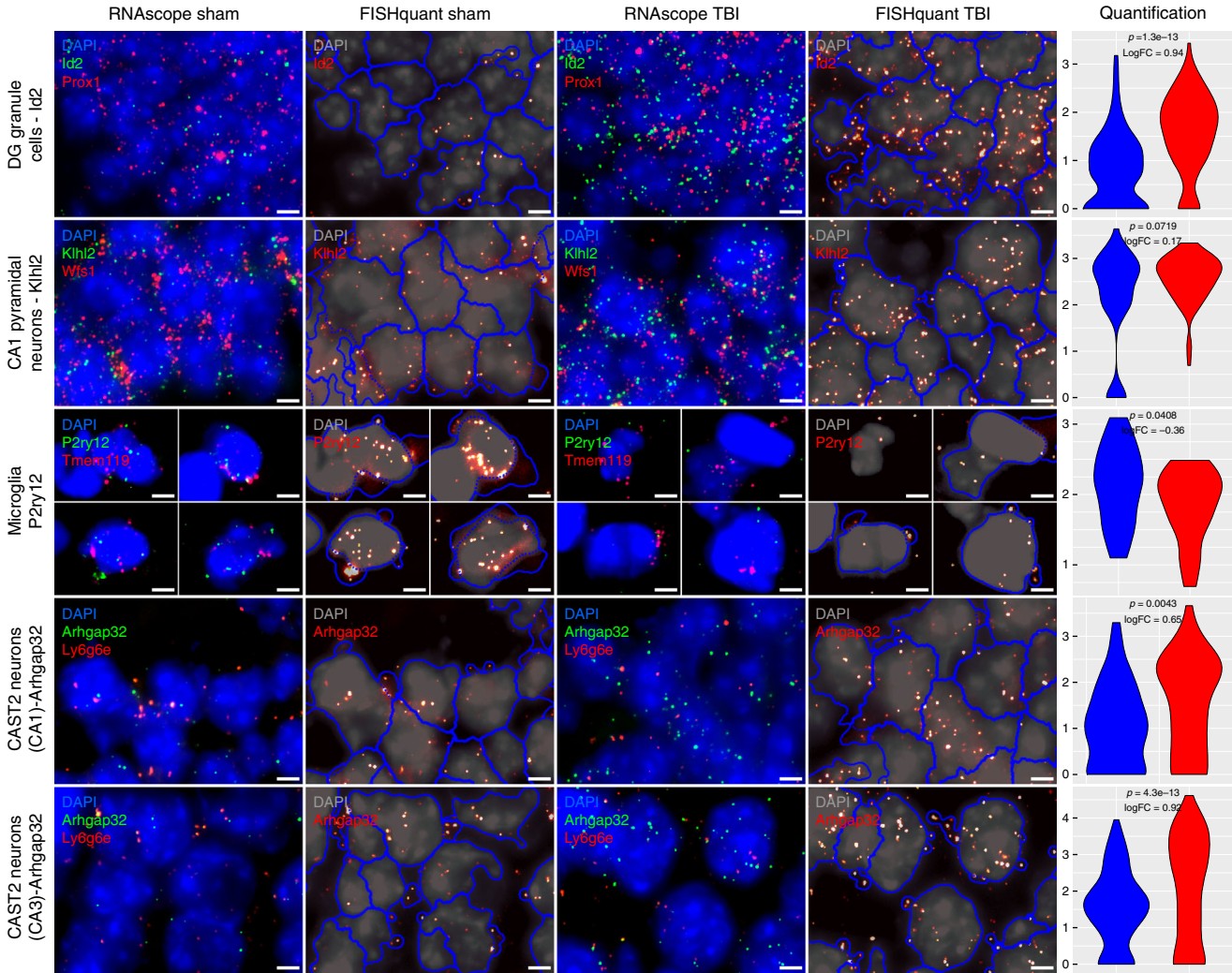

**Fig. 7** Validation of select cell type specific DEGs using RNAscope in situ hybridization. Representative fluorescent microphotographs for Sham and TBI showing each DEG of interest (selected from Fig. 6a) along with cell marker genes with DAPI in the background. The RNAscope images for Sham (first column) and TBI (third column) display cell colocalization of each DEG (green) and the corresponding cell marker gene (red). The corresponding FISHquant images for Sham (second column) and TBI (fourth column) show the automated cell segmentation (blue) overlaid on the 2D maximal projection of the DAPI z-stack images (gray) with the DEG of interest (green in the RNAscope images) indicated in bright dots. The quantification of the cell type specific DEGs is shown in the violin plots (5th column) with Sham in blue and TBI in red. Only cells which meet a certain count threshold for the cell type marker gene are considered the appropriate cell type and used in the DEG quantification (details in Methods). The ln(counts per cell) of the DEGs are shown on the y-axis with the p-value from a bimodal likelihood ratio test and log fold change (logFC) between TBI and Sham samples indicated. These figures have been zoomed in 5x from the original form to show high magnification detail of a few cells to more easily demonstrate colocalization of DEGs and cell markers. Lower magnification photos which indicate the region which was magnified are provided in Supplementary Fig. 11. Scale bars are 4 μm. Sample size was n = 8 mice/group

## Discussion

High-throughput parallel single cell sequencing analysis enabled us to characterize novel cell types, subtypes, and gene markers within the complex cytoarchitecture of the hippocampus under both physiological and pathological conditions, thus providing a rich resource for the neuroscience community to study hippocampus-related processes and diseases. In particular, our study unveiled key aspects of the hippocampal molecular and cellular adaptations during the acute phase of mTBI. Viewed holistically, our analyses of cell proportion changes, global transcriptome pattern shifts, differentially expressed genes, and perturbed pathways indicate that the majority of hippocampal cell types, including previously undefined cell populations, demonstrate various degrees of sensitivity to mTBI. As the transcriptome can instruct the functions of cells organized in circuits important

for processing of high order information, our results on the numerous alterations in cell-type specific genes and pathways and the reorganization of cell–cell gene co-expression patterns provide detailed molecular information crucial for our understanding of the mTBI pathology.

Our single cell information was also indispensable for the prioritization of numerous potential therapeutic pathways affected by mTBI in diverse cell types, including the thyroid hormone pathway as indicated by *Ttr*. The encouraging results from the use of T4 to normalize the disrupted pathways to improve cognitive behavior such as learning and memory support the potential of using single cell approaches to identify target pathways of therapeutic applications. It is important to note that such signals across multiple cell types can only be derived from studying all cell types individually as done in the current study.

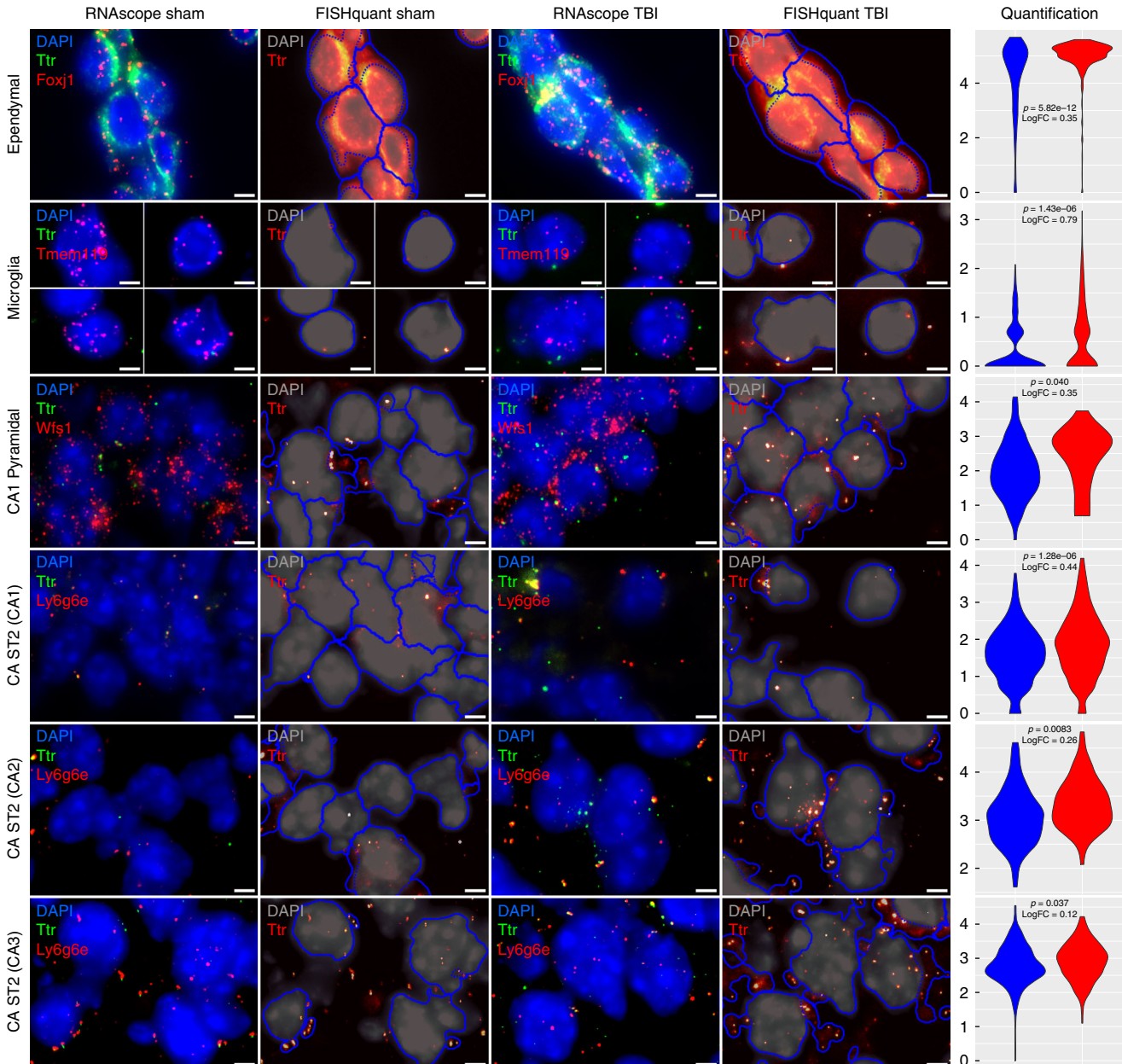

**Fig. 8** Validation of increased *Ttr* expression post-TBI using RNAscope in situ hybridization. Representative fluorescent microphotographs for Sham and TBI showing *Ttr* along with cell type markers with DAPI in the background. The RNAscope images for Sham (first column) and TBI (third column) display cell colocalization of *Ttr* (green) and the corresponding cell marker gene (red). The corresponding FISH-quant images for Sham (second column) and TBI (fourth column) show the automated cell segmentation (blue) overlaid on the 2D maximal projection of the DAPI z-stack images (gray) with *Ttr* (green in the RNAscope images) indicated in bright dots. The quantification of *Ttr* expression is shown in the violin plots (5th column) with Sham in blue and TBI in red. Only cells which meet a certain count threshold for the cell type marker gene are considered the appropriate cell type and used for *Ttr* quantification (details in Methods). The ln(counts per cell) of *Ttr* is shown on the *y*-axis with the *p*-value from a bimodal likelihood ratio test and log fold change (logFC) between TBI and Sham samples indicated. These figures have been zoomed in 5x from the original form to show high magnification detail of a few cells to more easily demonstrate colocalization between Ttr and cell type markers. Lower magnification photos which indicate the region which was magnified are provided in Supplementary Fig. 12. Scale bars are 4 µm. Sample size was $n = 8$ mice/group

The transcriptome perturbations by mTBI in individual cell types help pinpoint the cell origins of processes that likely guide mTBI pathogenesis, such as metabolic dysfunction in astrocytes and neurons and amyloid regulation involving ependymal and endothelial cells. These cell-level transcriptome patterns depict gene programs that may regulate and predict susceptibility to post-mTBI neurological disorders such as AD, PD, PTSD, neuroticism, and epilepsy. The information also has the potential to

guide treatments to improve mTBI outcome by targeting either specific cell types or broad cell interactions. For instance, the fact that genes involved in cell metabolism and amyloid processes were recurring findings across cell types and analytical approaches (cell-type specific DEGs, pan-hippocampal DEGs, pathways, and cell–cell gene co-expression) suggests that these are core processes in mTBI pathogenesis. Our single cell study opens new avenues to deconvolute the pathogenic processes involved in

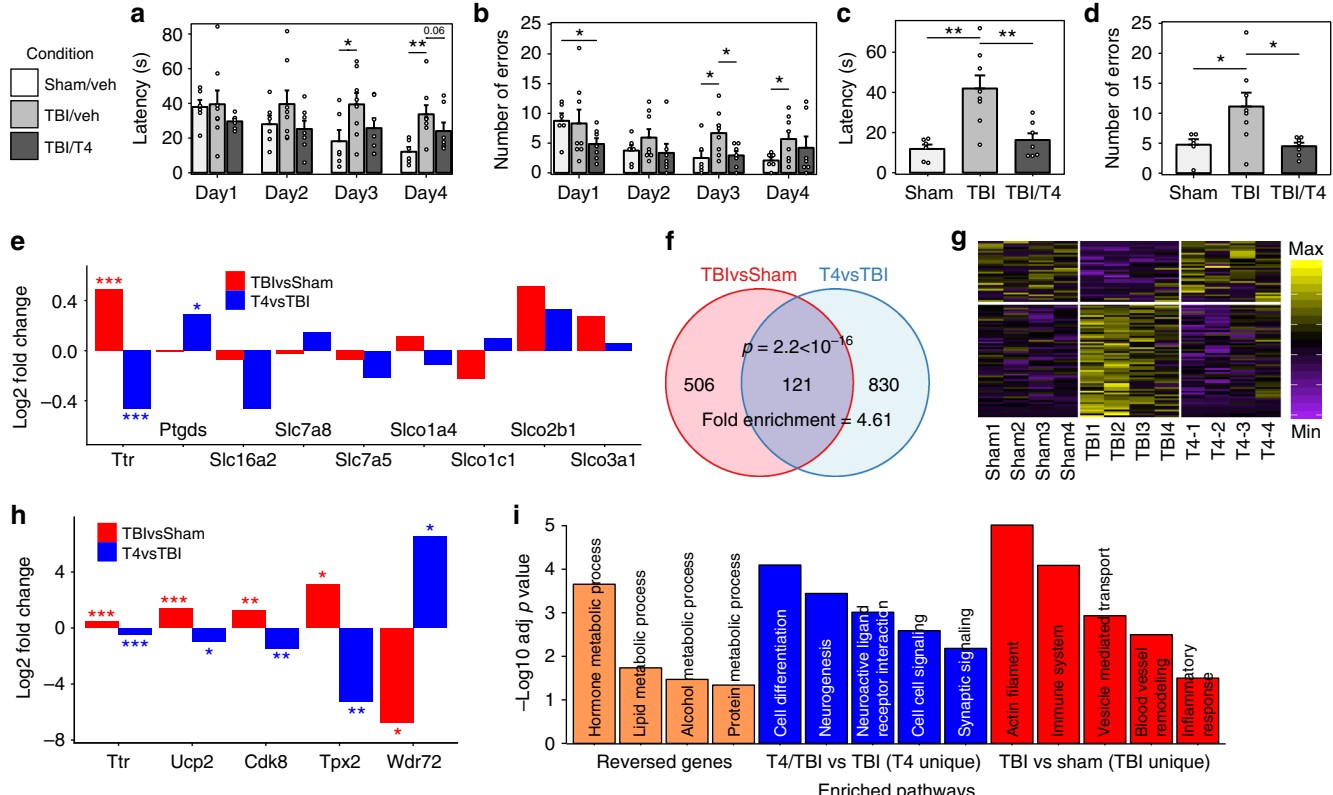

**Fig. 9** Validation of T4 treatment effects. **a, b** T4 treatment effects on learning through measurement of latency (**a**) and number of errors (**b**). **c, d** T4 treatment improves memory through measurement of latency (**c**) and number of errors (**d**) in Barnes Maze test. Bargraphs display mean values with error bars representing the standard error of the mean. Two-sided *t*-test was used to determine significance. **e** *Ttr* is the primary thyroid hormone transporter responsive to TBI and T4 treatment compared to other known transporters. **f–h** Gene expression profiles of T4 treatment experiments. **f** TBI and T4 treatment show significant overlap in DEGs. Significance in overlap was determined using Fisher's exact test. **g** Heatmap showing T4 reversed the expression patterns of 93 TBI-affected genes. **h** Examples of TBI genes reversed by T4. Significance of DEGs between the two conditions in **e** and **h** is determined by the negative binomial model, and the log2 fold change between two conditions based on the average gene expression values of the two groups was plotted on the *y*-axis. **i** Enriched pathways among the 93 TBI DEGs reversed by T4, pathways unique to T4 treatment (T4/TBIvsTBI) and those unique to TBI (TBIvsSham). Enrichment of pathways were determined using Fisher's exact test with Bonferroni corrected *p* values used for statistical significance. Adjusted \*$p < 0.05$, \*\*$p < 0.01$, \*\*\*$p < 0.001$. Error bars are s.e.m. Sample size: **a–d** Sham/Veh: $n = 6$, TBI/Veh: $n = 8$, TBI/T4: $n = 7$; **e, h** $n = 4$/group

mTBI in individual brain cell populations and prioritize both cell-type specific (such as *Arhgap32* in neurons and *Apoe* in ependymal cells) and potential broad-spectrum pan-hippocampal targets (such as transthyretin and humanin).

Concussive brain injury is the most common form of brain injury in sports, domestic, and military settings and has been associated with numerous long-lasting and debilitating neurological consequences. The identified genes, processes, and cell types vulnerable to acute concussive injury will form the foundation for mechanistic studies and for the development of novel therapeutic strategies for mTBI and related neurological disorders. The cell type-specific signals may also facilitate future development of molecular diagnostic markers considering difficulties to diagnose concussive injury using conventional neuroimaging examinations[3]. Future in-depth studies to test the functionality as well as the potential diagnostic and therapeutic values of the predicted novel cell types, cell type specific mTBI targets, and cell-cell coordination are warranted.

## Methods

**Animals and mild fluid percussion injury (FPI).** Male C57BL/6 J (B6) mice of 10 weeks of age (Jackson Laboratory, Bar Harbor, ME, USA) weighing between 20 and 25 g were group housed in cages ($n = 3$–4/group) and maintained in environmentally controlled rooms (22–24 °C) with a 12 h light/dark cycle. Mice were randomized to receive either FPI or Sham surgeries, with no investigator blinding. FPI was performed with the aid of a microscope[55] (Wild, Heerburg, Switzerland), where a 1.5-mm diameter craniotomy was made 2.5 mm posterior to the bregma and 2.0 mm lateral (left) of the midline with a high-speed drill (Dremel, Racine, WI, USA). A plastic injury cap was placed over the craniotomy with silicone adhesive and dental cement. When the dental cement hardened, the cap was filled with 0.9% saline solution. Anesthesia was discontinued, and the injury cap was attached to the fluid percussion device. At the first sign of hind-limb withdrawal to a paw pinch, a mild fluid percussion pulse (1.5–1.7 atm, wake up time greater than 5 min) was administered. Sham animals underwent an identical preparation with the exception of the lesion. Immediately following response to a paw pinch, anesthesia was restored and the skull was sutured. Neomycin was applied on the suture and the mice were placed in a heated recovery chamber for approximately an hour before being returned to their cages. After 24 h, mice were sacrificed and fresh hippocampal tissue was dissected for use in Drop-seq ($n = 3$/group with one animal per group per day; sample size was determined based on previous single cell studies that demonstrated sufficient statistical power). All experiments were performed in accordance with the United States National Institutes of Health Guide for the Care and Use of Laboratory Animals and were approved by the University of California at Los Angeles Chancellor's Animal Research Committee.

**Tissue dissociation for Drop-seq.** The protocol by Brewer et al[56]. was used to suspend cells at a final concentration of 100 cells/μl in 0.01% BSA-PBS by digesting freshly dissected hippocampus tissue with papain (Worthington, Lakewood, NJ, USA). Briefly, hippocampi were rapidly dissected from the ipsilateral side of the

brain on ice. The hippocampi were transferred into 4 ml HABG (Fisher Scientific, Hampton, NH, USA) and incubated in water bath at 30 °C for 8 min. The supernatant was discarded and the remaining tissue was incubated with papain (12 mg in 6 ml HA-Ca) at 30 °C for 30 min. After incubation, the papain solution was removed from the tissue and washed with HABG three times. Using a siliconized 9-in Pasteur pipette with a fire-polished tip, the solution was triturated approximately ten times in 45 s. Next, the cell suspension was carefully applied to the top of the prepared OptiPrep density gradient (Sigma Aldrich, St. Louis, MO, USA) and floated on top of the gradient. The gradient was then centrifuged at 800 g for 15 min at 22 °C. We aspirated the top 6 ml containing cellular debris. To dilute the gradient material, we mixed the desired cell fractions with 5 ml HABG. The cell suspension containing the desired cell fractions was centrifuged for 3 min at 22 °C at 200 g, and the supernatant containing the debris was discarded. Finally, the cell pellet was loosened by flicking the tube and the cells were re-suspended in 1 ml 0.01% BSA (in PBS). This final cell suspension solution was passed through a 40-micron strainer (Fisher Scientific, Hampton, NH, USA) to discard debris, followed by cell counting.

**Drop-seq single cell barcoding and library preparation**. Barcoded single cells, or STAMPs (single-cell transcriptomes attached to microparticles), and cDNA libraries were generated following the drop seq protocol from Macosko et al[12]. and version 3.1 of the online Drop-seq protocol [http://mccarrolllab.com/download/905/]. Briefly, single cell suspensions at 100 cells/μl, EvaGreen droplet generation oil (BIO-RAD, Hercules, CA, USA), and ChemGenes barcoded microparticles (ChemGenes, Wilmington, MA, USA) were co-flowed through a FlowJEM aquapel-treated Drop-seq microfluidic device (FlowJEM, Toronto, Canada) at recommended flow speeds (oil: 15,000 μl/hr, cells: 4000 μl/hr, and beads 4000 μl/hr) to generate STAMPs. The following modifications were made to the online published protocol to obtain enough cDNA as quantified by a high sensitivity BioAnalyzer (Agilent, Santa Clara, CA, USA) to continue the protocol: (1) The number of beads in a single PCR tube was 4000. (2) The number of PCR cycles was 4 + 11 cycles. (3) Multiple PCR tubes were pooled. The libraries were then checked on a BioAnalyzer high sensitivity chip (Agilent, Santa Clara, CA, USA) for library quality, average size, and concentration estimation. The samples were then tagmented using the Nextera DNA Library Preparation kit (Illumina, San Diego, CA, USA) and multiplex indices were added. After another round of PCR, the samples were checked on a BioAnalyzer high sensitivity chip for library quality check before sequencing. A cell doublet rate of 5.6% was obtained by running the microfluidic device without the lysis buffer and counting the percentage of cell doublets through three separate runs.

**Illumina high-throughput sequencing of Drop-seq libraries**. The Drop-seq library molar concentration was quantified by Qubit Fluorometric Quantitation (ThermoFisher, Canoga Park, CA, USA) and library fragment length was estimated using a Bioanalyzer. Sequencing was performed on an Illumina HiSeq 2500 (Illumina, San Diego, CA, USA) instrument using the Drop-seq custom read 1B primer (GCCTGTCCGCGGAAGCAGTGGTATCAACGCAGAGTAC) (IDT, Coralville, IA, USA). Paired end reads were generated using custom read lengths of 24 for read 1 and 76 for read 2 and an 8 bp index read for multiplexing. Read 1 consists of the 12 bp cell barcode, followed by the 8 bp UMI, and the last 4 bp on the read are not used. Read 2 contains the single cell transcripts.

**Drop-seq data pre-processing and quality control**. The fastq files of the Drop-seq sequencing data were processed to digital expression gene matrices using Drop-seq tools version 1.12 [http://mccarrolllab.com/download/922/]. The protocol outlined in the Drop-seq alignment cookbook v1.2 [http://mccarrolllab.com/wp-content/uploads/2016/03/Drop-seqAlignmentCookbookv1.2Jan2016.pdf] was followed, using default parameters. Fastq files were converted to BAM format and cell and molecular barcodes were tagged. Reads corresponding to low quality barcodes were removed. Next, any occurrence of the SMART adapter sequence or polyA tails found in the reads was trimmed. These cleaned reads were converted back to fastq format to be aligned to the mouse reference genome mm10 using STAR-2.5.0c. After the reads were aligned, the reads which overlapped with exons were tagged using a RefFlat annotation file of mm10. A percentage of the Chemgenes barcoded beads which contain the UMIs and cell barcodes were anticipated to have synthesis errors. We used the Drop-seq Tools function DetectBeadSynthesisErrors to quantify the Chemgenes beads batch quality and estimated a bead synthesis error rate of 5–10%, within the acceptable range. Finally, a digital gene expression matrix for each sample was generated where each row is the read count of a gene and each column is a unique single cell. The transcript counts of each cell were normalized by the total number of UMIs for that cell. These values are then multiplied by 10,000 and log transformed. Digital gene expression matrices from the six samples (3 Sham and 3 TBI samples) were combined to create three different pooled digital gene expression matrices for: (1) all Sham samples, (2) all TBI samples and (3) combined Sham and TBI samples. Single cells were identified from background noise by using a threshold of at least 500 genes and 900 transcripts.

**Identification of cell clusters**. The Seurat R package version 1.4.0.1 [https://github.com/satijalab/seurat] was used to project all sequenced cells onto two

dimensions using t-SNE and density-based spatial clustering (DBSCAN) was used to assign clusters. To further refine the neuronal cell clusters, the BackSPIN software[14] was used to perform biclustering of the single cells identified to be neuronal cells to further resolve this cell type. Biclustering has been previously demonstrated to differentiate between cell types which cannot be captured by traditional t-SNE-based approaches[14]. BackSPIN was run with default parameters, selecting for the top 2000 most highly variable genes and proceeding with 5 levels of biclustering.

**Identification of marker genes of individual cell clusters**. We defined cell cluster specific marker genes from our Drop-seq dataset using a bimodal likelihood ratio test[57]. To determine the marker genes, the single cells were split into two groups for each test: the cell type of interest and all remaining single cells. To be considered in the analysis, the gene had to be expressed in at least 30% of the single cells from one of the groups and there had to be at least a 0.25 log fold change in gene expression between the groups. Multiple testing was corrected using the Benjamini–Hochberg method and genes with an FDR < 0.05 are defined as marker genes. We explored the gene-gene correlation within-group and between-group for each cell type and confirmed the consistency of cell type identification between samples (Supplementary Fig. 1).

**Resolving cell identities of the cell clusters**. We use two methods to resolve the identities of the cell clusters. First, known cell-type specific markers from previous studies were curated and checked for expression patterns in the cell clusters. A cluster showing high expression levels of a known marker gene specific for a particular cell type was considered to carry the identity of that cell type. Second, we evaluated the overlap between known marker genes of various cell types with the marker genes identified in our cell clusters. Overlap was assessed using a Fisher's exact test and significance was set to Bonferroni-corrected $p < 0.05$. A cluster was considered to carry the identity of a cell type if the cluster marker genes showed significant overlap with known markers of that cell type. The two methods showed consistency in cell identity determination.

Known markers for major hippocampal cell types and neuronal subtypes were retrieved from Zeisel et al.[14], Habib et al.[15], and the Allen Brain Atlas[58]. These markers were sufficient to define all major cell types as well as GABAergic neurons, dentate gyrus (DG), CA1 and CA3 pyramidal neurons.

**Quantitative assessment of global transcriptome shifts**. For each cell type, we generate two representative cells, one for the Sham group and the other for mTBI condition by calculating the average gene expression of each gene for each group within that cell type. We then calculate the Euclidian distance in gene expression between these representative cells as a metric to quantify the effect of TBI on each cell type. To determine if the observed Euclidian distance between Sham and mTBI cells within each cell type is significantly larger than that of random cells, we estimated a null distribution by calculating the Euclidian distance between randomly sampled cells of the given cell type. This permutation approach is repeated for a total of 1000 times to generate the null distribution, which is compared to the Euclidian distance generated from the true TBI and Sham groups to determine an empirical p value. To correct for multiple testing across all the cell types tested, we applied a Bonferroni correction to retrieve adjusted p values.

**Cell–cell gene co-expression analysis**. We assessed cell–cell gene co-expression based on gene-level correlation patterns between any two given cell types (Fig. 6a). To infer directionality of the interactions between two cell types, we defined a cell type whose marker genes encode secreted peptides based on Uniprot information as the source cell type, and then correlated the peptide-encoding marker genes from the source cell type with genes in the target cell type. To deal with the sparse nature of single cell data, we averaged the gene expression of Sham and TBI samples respectively for each cell type. An interaction score is calculated from the sum of the correlation p-values for each peptide, assuming that a peptide from a source cell type with strong correlations with many genes in the target cell type would have a high score and indicate strong interactions. To determine the significance of interaction, we use a permutation approach in which a null distribution is drawn from the interactions scores generated by the correlations between source peptides and target genes where the expression values for each target gene has been shuffled independently. The genes correlated with each peptide were tested for pathway enrichment in KEGG, Reactome, BIOCARTA, GO Molecular Functions, and GO Biological Processes to infer the key pathways involved in the interactions.

To assess the validity of our method, we benchmarked how well our gene coexpression analysis could recapitulate the known cell–cell interactions in the hippocampal trisynaptic circuit. As the neuronal cell types involved in this neural circuit and present in our data are glutamatergic neurons (DG granule cells, CA1 pyramidal neurons, and CA3 pyramidal neurons) we used genes involved in glutamate secretion to define source cell types and used the same scoring and permutation approach above to determine significance of gene coexpression between cell types. Due to the very small numbers of CA1 and CA3 cells in our dataset hence limited power, we utilized data from a previous DroNc-seq study[59] involving much larger populations of CA1 pyramidal cells, CA3 pyramidal cells, and DG granule cells from 6 control animals.

**Identification of DEGs between Sham and TBI**. Within each identified cell type, Sham and TBI samples are compared for differential gene expression using a bimodal likelihood ratio test[57]. To be considered in the analysis, the gene had to be expressed in at least 30% of the single cells from one of the two groups for that cell type and there had to be at least a 0.25 log fold change in gene expression between the groups. We correct for multiple testing using the Benjamini–Hochberg method and genes with an FDR < 0.05 were used in downstream pathway enrichment analyses (unless explicitly noted that a p-value of 0.01 was used instead to retrieve suggestive pathways). Enrichment of pathways from KEGG, Reactome, BIO-CARTA, GO Molecular Functions, and GO Biological Processes was assessed with Fisher's exact test, followed by multiple testing correction with the Benjamini–Hochberg method.

**Comparison of single cell vs. bulk tissue DEGs**. To define the advantages gained by employing single cell sequencing, we simulated bulk tissue gene expression by averaging the gene expression across all TBI single cells and all Sham single cells for each animal. These in silico bulk tissue Sham and TBI samples were then compared for differential gene expression using a bimodal likelihood ratio test[57]. FDR < 5% was used as the cutoff to determine tissue-level DEGs, which were then compared against those from the single-cell analysis.

**RNAscope in situ hybridization**. We used RNAscope Multiplex in situ hybridization (Advanced Cell Diagnostics) to assess the expression of the genes of interest (Wang, J Molec Diag, 2012), according to manufacturer user manual for fresh frozen tissue. For each gene, n = 8/group was used. Two fresh frozen brain slices from each hippocampus (15 μm) were mounted on glass slides in 4% neutral buffered paraformaldehyde (Fisher Scientific) for 15 min at 4 °C. We rinsed the slides twice in PBS, dehydrated them in 50, 70, 100, and 100% ethanol, and stored slices in fresh 100% ethanol for overnight at −20 °C. Slides were dried at room temperature for 5 min. We drew a hydrophobic barrier on slides around the brain slices to avoid the spreading of solutions, and then treated the slides with protease IV at room temperature for 30 min followed by a PBS wash. We applied probes to co-localize the target gene and a cell marker gene in the same tissue to the slides and incubated them at 40 °C for 2 h in the HybEZ oven. Each RNAscope target probe contains a mixture of multiple ZZ oligonucleotide probes that are bound to the target RNA: Ttr probe (GenBank accession number NM_013697.5; target region 141–1149, Arhgap32 probe (GenBank accession number NM_001195632.1; target nt region, 1880–2864), Id2 probe (GenBank accession number NM_010496.3; target nt region, 57–794), Foxj1 probe (GenBank accession number NM_008240.3; target nt region, 1121–1836); Ly6g6e probe (GenBank accession number NM_027366.1; target nt region, 2–902), Prox1 probe (GenBank accession number NM_008937.2; target nt region, 590–1769), Tmem119 probe (GenBank accession number NM_146162.2; target nt region, 2–1106), Wfs1 probe (GenBank accession number NM_011716.2; target nt region, 757–1629), Klhl2 probe (GenBank accession number NM_178633.3; target nt region, 534–1477), P2ry12 (GenBank accession number NM_027571.3; target nt region, 739–1854). The negative control sections received RNase treatment before performing the RNAscope assay, and the positive control contained a housekeeping RNA (RTU mixture of three probes targeting POLR2A in channel C1, PPIB in channel C2, and UBC in channel C3). We incubated the slides with amplifier probes (AMP1, 40 °C for 30 min; AMP2, 40 °C for 15 min; AMP3, 40 °C for 30 min). We then incubated the slides with fluorescently labeled probes by selecting a specific combination of colors associated with each channel: green (Alexa 488 nm), orange (Alexa 550 nm), and far red (Alexa 647 nm). We used AMP4 FL Alt A to detect genes of interest. We washed the slides with 1× washing buffer twice in between incubations. After air drying the slides we coverslipped them with a ProLong Gold Antifade mounting medium containing DAPI (Invitrogen). Fluorescent images were captured at 400× magnification (Zeiss Imager.Z1; Gottingen, DE) using the Axiovision software (Carl Zeiss Vision, version 4.6). According to manufacturer, each dot represents a single molecule of mRNA.

**RNAscope quantification**. RNAscope ISH was quantified using the automated FISH-quant software[44]. Briefly, each stained tissue section was imaged in a z-stack of 5 images with 555 nm of distance between each image. Each set of 5 z-stack images for DAPI, Alexa 488 nm, Alexa 550 nm, and/or Alexa 647 nm depending on the sets of genes being multiplexed were collapsed into a single .tif file for each channel which were used in the FISH-quant software with the following settings: pixel size—555 nm (x, y, and z), refractive index—1.0002, numeric aperture—0.75, 554, and 576 excitation and emission wavelength (Alexa 550 nm), 501 and 523 excitation and emission wavelength (Alexa 488 nm). Prior to cell segmentation, a 2D maximal projection was made from each image z-stack using a TENG focus operator for DAPI and a HELM focus operator for Alexa 550 nm and Alexa 488 nm images with global focus measurement. These 2D projections were used in the CellProfiler[43] software to define nuclei and cell segmentations. Using these outlines in FISH-quant, preliminary parameters for each gene were determined from a set of 3 representative images. These parameters were applied in batch-mode to quantify all images, and optimal threshold parameters for determining true mRNA spots from background were defined based on distributions for: sigma-xy, sigma-z, and the amplitude of the RNA spots. To ensure quantification of a DEG in cell types of interest, we consider only cells which meet a certain RNA spot threshold of the cell type specific marker gene to be the appropriate cell type. This threshold is determined by looking for the knee of gene counts per cell. The following thresholds were used: CA1 Pyramidal cells—6, Microglia—11, Ependymal—15, CA Subtype 2–8, Dentate Gyrus granule cells—20. Within the group of cells which pass this threshold, we quantify the counts per cell of the DEG of interest and compare the count distributions between TBI and Sham groups using a bimodal likelihood ratio test[57].

**T4 treatment**. L-Thyroxine sodium salt pentahydrate (T4, Sigma Chemical Co., St. Louis, MO, USA) dissolved in saline vehicle (154 nM NaCl) was injected i.p. twice at 1 and 6 h after FPI in the treatment group (n = 6 mice) at 1.2 μg/100 g body weight. Control FPI mice (n = 6) received vehicle (saline).

**Behavioral tests for T4 treatment experiments**. Mice from the Sham, TBI, and T4 treatment groups were tested on the Barnes maze 7 days after injury to assess learning acquisition and memory retention[60]. For learning, animals were trained with two trials per day for 4 consecutive days, and memory retention was assessed two days after the last learning trial. The maze was manufactured from acrylic plastic to form a disk 1.5 cm thick and 120 cm in diameter, with 40 evenly spaced 5 cm holes at its edges. The disk was brightly illuminated (900 lumens) by four overhead halogen lamps to provide an aversive stimulus to search for a dark escape chamber hidden underneath a hole positioned around the perimeter of a disk. All trials were recorded simultaneously by a video camera installed directly overhead at the center of the maze. A trial was started by placing the animal in the center of the maze covered under a cylindrical start chamber; after a 10 s delay, the start chamber was raised. A training session ended after the animal had entered the escape chamber or when a pre-determined time (5 min) had elapsed, whichever came first. All surfaces were routinely cleaned before and after each trial to eliminate possible olfactory cues from preceding animals. After the memory test the animals were sacrificed immediately by decapitation and the fresh hippocampal tissues were dissected out, frozen in liquid nitrogen, and stored in −80 °C for bulk tissue RNA-sequencing.

**RNA-seq analysis of T4 treatment experiments**. After completion of the behavior test (11 days post-injury/treatment), hippocampal tissues were dissected from the Sham, untreated TBI, and T4-treated TBI animals (n = 6/group for behavior, and n = 4/group selected for sequencing). QuantSeq 3′ libraries were prepared from cDNA samples using the QuantSeq 3′ mRNA-Seq Library Prep Kit (Lexogen, Greenland, NH, USA). Libraries were run on a HiSeq4000. Reads were aligned with STAR and read counts per gene were generated using the BlueBee platform. Differentially expressed genes between the different groups (Sham, TBI untreated, and TBI T4 treatment) were determined using negative binomial models[61]. DEGs with p < 0.05 were included in gene signatures which were checked for pathway enrichment. DEGs between TBI and Sham were compared for overlap with DEGs between T4 treated mice and untreated TBI mice.

**Code availability**. Our analytical workflow was stored and freely accessible as a SnakeMake file [https://github.com/darneson/DropSeq].

## Data availability
The sequence data that support the findings of this study have been deposited in the Gene Expression Omnibus repository, with the series record GSE101901.

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

## Acknowledgements

We thank Dr. Weizhe Hong for advice on neuronal subpopulation analysis and helpful discussions. X.Y. and F.G-P. are funded by R01 DK104363 and R21 NS103088. F.GP. is funded by R01 NS50465 and UCLA BIRC. D.A. is funded by Hyde Fellowship and NIH-NCI National Cancer Institute T32CA201160.

## Author contributions

D.A. participated in the experimental design, collected and analyzed sequencing datasets, and wrote the paper. Y.Z., H.R.B., I.S.A., Z.Y., G.Z conducted animal, Drop-seq, and

immunohistochemistry experiments, and edited the manuscript. X.Y. and F.G-P. conceived the study, designed and coordinated the study, and wrote the manuscript.

## Additional information

**Competing interests:** The authors declare no competing interests.

