## [Peer Review File · Nature Communications]

Reviewers' comments:

Reviewer #1 (Remarks to the Author):

This manuscript is mostly well written, the authors use a cutting edge unbiased technique (Drop-sequencing) to determine the sequencing of thousands of cells from the hippocampus of animals that received a concussive injury. Using Drop-seq they categorize cells according to their similarities in gene expression patterns and extract cell type specific genetic markers affected by mild concussive injury. The authors were able to uncover hippocampal cell types most sensitive to the specific animal model used (concussive mild TBI (mTBI)) as well as the vulnerable genes, pathways and cell-cell interactions predictive of disease pathogenesis in a cell-type specific manner. As a proof of concept the authors targeted a specific (Ttr) pathway.

There is a lot of information that the study brings to the field with many possibilities for testing targets however the implications they draw from their findings are overstretched. The utility in their findings are helpful for understanding the possible contributions of cellular populations and how they can respond differently to the same injury (at the single cell level). Although the authors address the "flaws/downsides" of the Drop Seq method and the samples used, the continuous texts of how their work has large implications masks this information. There are numerous points that should be clarified:

Major points:

In general the authors are drawing strong conclusions from only having analyzed correlations and this is a major problem.

LN 158: How do traditional morphology-based approaches not inform on function whereas genomic features does? saying these genomic features determine function is overreaching. In the first section of the Results, the conclusion is that the Drop-seq approach is able to identify novel genes specific to certain cell populations which are validated by matching known gene expressions and anatomical location. Likewise, Drop-seq further differentiates populations that might seem morphologically or genetically (by previous methods) similar but now differentially cluster. There are no functional assays suggesting differences in, for example, the two CA Subtype neurons.

Fig 1: The authors mention later in the paper that TBI samples were enriched for Ependymal cells. Given that astrogliosis and microglia proliferation occurs in response to injury, how do the other cell populations compare between Sham and TBI?

LN 161: They only used changes in cell numbers to identify vulnerability of cells but this could be a confounding. They could have supplemented by looking at genomic changes.

LN 108: How do they know that in other regions the targets mentioned are also cell-specific?

LN 139: How did they confirm the unannotated clusters are neuronal populations?

Line 172-175: While oligodendrocytes, astrocytes, and microglia do appear weakly shifted in Fig1a, it is difficult to say with certainty that the claim holds true for CA1 neurons and CA subtype2 neurons. Do the authors have a more quantitative measure to support their claim? In contrast, CA Subtype1 looks to be more differentially clustered but the authors do not include the population in their list. Why?

LN 194: it is incorrect to say that "this could be a novel property of hippocampal cells"; it could happen between cell types of other regions as well.

LN 250: If they (DEGs) are masked in bulk, can they be strong drivers? (The resolution of each technique makes this a point of discussion.)

Figure 7A: It would be important to validate that Ttr is upregulated by TBI. While the authors attempt to show this through immunohistochemistry at the CA regions, the staining is not conclusive or quantified.

Figure 7 B and C behavior results: On Day1, TBI/T4 mice learn faster than Sham, but they don't show progressive learning as shams, can they still say T4 protects learning?

There are no data to show velocity of mice during Barnes maze, the shortened latency of the TBI/T4 group could be the result of higher moving speed. They need to show velocity, and number of errors to conclusively state that this is an effect in learning and memory.

Figure 7 D, G and H. The authors chose to analyze overlapping genes between TBI vs Sham and TBI/T4vsTBI, if TBI causes metabolism abnormalities, then they treat mice with a metabolism simulating reagent, of course the enrichment analysis will show mainly metabolism related GO terms. This result is heavily biased by their analysis protocol. It would be more informative to compare all altered genes in T4vsTBI.

Cell-cell interaction section: This section is difficult to understand for its implications. The gene co-expression method does give some insight to how genes of one cell type correlates with genes of another. However, the analysis does not provide direct information about how cell-cell interactions are disrupted. At best, the methodology here can conclude a shift in cell-cell interaction in the injured brain, but cannot actually inform which cell communication pathways are disrupted. To address this at a minimum the authors need to change the title so it does not imply a downshift in cell-cell interaction. This is especially true since the authors state both increase in gene co-expressions (astrocyte and ependymal to neurons, microglia to oligodendrocytes) as well as decreases (microglia to neurons, oligodendrocytes to neurons).

Given the authors' findings about the ability to detect cell-type-specific genes for treatment, what was the rationale for choosing Ttr which the authors acknowledge is a pan-hippocampal target? This is especially notable given that the authors specifically highlight the effect of TBI on DG Granule cell gene expressions (Fig 1a and Fig3a). Furthermore, does targeting Ttr with T4 treatment effect certain cell types more or is it effective across all cell types? The data only looks at pan-hippocampal sequencing data.

Methods:

RNA-seq analysis of T4 treatment experiments: Authors should write whether the tissues collected for RNA-seq were from the mice that ran the behavioral experiment. The n don't line up (n=6/group for behavior, n=4/group for RNA-seq). Furthermore, what time point were the animals taken down for RNA-seq (if after behavior, it would be at 11 days post-injury and treatment; if sooner, how long after the last injection of T4)?

Minor points:

Line 57: what "spectrum of the hippocampus" refers to in line 57.

LN 131: Example of a "reachy statement."

Line 153-155: Has damage to subiculum been explicitly linked to TBI models?

LN 169: Confusing text.

LN 212-216: Not new information to the field.

LN 254: what is that value? Don't state it clearly.

Line 276: Please provide a reference for the line referring to a link between mTBI and increased tendency for neuroticism post-TBI.

Figure 6: needs a figure legend (color).

Figure 7E and F: error bars for these plots should be shown.

Line 374 – 378: The sentence is difficult to parse. Rewrite it for clarity.

Reviewer #2 (Remarks to the Author):

In this study, Yang and colleagues investigated the cell types and molecular pathways in the hippocampus that were altered by concussive brain injury by applying single-cell RNA sequencing to a mouse model. They compared the single-cell expression profiles of hippocampus between three mTBI and three control animals. Identify the vulnerable cell types, and specific pathways that correlated with the injury. They further prioritized potential molecular targets for therapeutic interventions based on differentially expression genes, and experimentally demonstrated that the Ttr was highly expressed post injury, potentially as a compensatory mechanism for thyroxine T4, and the injection of T4 post-mTBI can reversed the transcriptional changes induced by mTBI.

Overall, this is a timely study of the molecular mechanisms underlying mTBI using a cutting-edge single-cell transcriptional profiling approach. It revealed novel insights related to the cell types and pathways specific to mTBI. The findings provide new targets for therapeutic intervention.

One major criticism of this study is a general lack of experimental validation on the findings by single-cell transcriptional profiling. The authors did presented many ISH images, including many in Figure 2. However, these were the existing single-gene low-resolution ISH data produced by the Allen Brain Institute, on normal mouse brain. Multiplex RNA in situ data on the mTBI and sham animal are necessary to validate some of the key findings in this study.

Second, some of the analyses were a bit loose. For example, the analysis of cell-cell interaction in Page 10 is really interesting, but how robust is the method used for the analysis? Cell-cell interactions can't be studied without the context of spatial organization. An observation that cell A is secreting some peptides and cell B has the surface receptors of such peptides does not necessarily mean real interaction unless there is evidence that the peptides can reach cell B within the 3D space of hippocampus.

Reviewer #3 (Remarks to the Author):

This paper by Arneson and colleagues reports the results of a high throughput single cell sequencing study performed on thousands of individual hippocampal cells isolated from the mouse brain after concussive mild TBI. The study also identifies the transthyretin (Ttr) gene, a gene coding for a transport protein that carries the T4 thyroid hormone across the blood-brain barrier, and validates T4 as a novel treatment for mTBI. Overall, the manuscript is well-written and the data are novel, convincing and of potential high impact to the neuroscience field. The experimental flow of the paper follows a straightforward and logical path to acquire some interesting findings. However I do have some relatively minor issues that should be taken into consideration in a revised manuscript.

- The authors list in the introduction a number of critical longstanding questions that they claim their study will help resolve. One of these questions regards the identity of the hippocampal cell types that are the most vulnerable to concussive mild TBI. However, based on data presented in Fig. 1 and Table 1, it can be assumed that basically all of the hippocampal cell types are vulnerable to mTBI. Were some of the differentially-expressed genes enriched for pathways involved in cell death? Along the same line, the abstract should be more informative on this matter.

- Another goal of the study was to identify diagnostic and prognostic biomarkers for mTBI. Among the biomarkers that were identified by the unbiased single-cell sequencing (Drop-Seq) method are Id2, p2ry12, apoe, and Itm2a. What is unclear, however, is whether any of these biomarkers can be measured other than by post-mortem brain examination. Adding a confirmation of upregulation of protein expression for one of these biomarkers in the cerebrospinal fluid or blood of mTBI mice would be a significant plus value to the study.

- The circos plots included in Fig. 4b-c depict genes coding for secreted proteins that have the potential to interact with receptor-encoding genes in target cells during mTBI pathogenesis. This dataset is critical to the overall importance of the work, and also for future studies that could derive from it, but should be improved in a number of ways: 1) the lines are just too hard to follow from one cell type to another, 2) each secreted factor should be linked to an identifiable specific receptor in the target cell, and the name of that receptor should appear in the revised figure.

- Figure 7a: The immunofluorescence staining for Ttr is not convincing (weak signal and some images are out of focus). Also, Fig. 7 and Supplementary Fig. 10 are somewhat

redundant.

- Figure 7b: Did the T4 treatment correct TBI-induced learning deficit or not? Significance is not shown between the TBI/Veh and TBI/T4 groups in the graph.

Response to Reviewer Comments

We thank the Editor and the three Reviewers for the constructive and thoughtful comments to help us improve the study. We have revised the manuscript to address each of the comments, as outlined below.

Editor's comments:

E1. Need further convincing data to follow-up on the current correlational observations made about distinct cell types and their role in TBI, a point raised by all referees. We appreciate the data on T4 amelioration of TBI, but would agree with Reviewer #1's assessment, in which he/she asked for the rationale of pursuing *Ttr* as a hit even though it is not cell type specific. Therefore we'd like to see further follow-up on cell type specific observations.

Response: We appreciate the Editor and Reviewers' suggestions on adding additional rationale and validations to take our Drop-seq studies beyond the correlative level. Indeed, our T4 experiment was designed as such an effort. We have further substantiated the rationale for choosing *Ttr* as a proof of concept to validate the applicability of single cell information for the design of therapeutic interventions (Page 17). The idea behind choosing *Ttr* was mainly because it was upregulated by TBI in each of the individual hippocampal cell types detected in our study, making it a very comprehensive and robust TBI target. Without the cell-type specific analysis for individual cell types, we would not have been able to prioritize *Ttr* based on this unique feature. We believe targets like *Ttr* have the capacity to mediate the broad effects of TBI on the hippocampus. As such, modifying *Ttr* activities will help achieve stronger therapeutic effects than modulating a cell-type specific target, which likely can only normalize a limited aspect of hippocampal functionality and may not have strong functional effects. Our T4 treatment results strongly support the validity of our rationale and confirm that *Ttr* is not just correlated with TBI but plays a causal role in the TBI pathology.

We have taken suggestions by the Reviewers and used multiplex RNA ISH to validate select cell type specific genes highlighted in Figure 6a, which were predicted by Drop-seq to be altered by TBI. We used the RNAscope ISH to confirm the predicted cell-type specific expression as well as the direction of alteration by TBI in the predicted cell types (new Figure 7). In parallel, we also used RNAscope ISH to test *Ttr* expression in multiple hippocampal regions and demonstrated its upregulation in various cell types. Therefore, these newly added multiplex ISH experiments, coupled with the T4 treatment experiment targeting *Ttr*, validated both cell-type specific and pan-hippocampal targets of TBI and brought correlation closer to causality. To our knowledge, our study is the first example to utilize the quantitative single cell information to prioritize targets and to demonstrate therapeutic effects. We believe this is a highly innovative and significant aspect of our study compared to other single cell studies.

E2. Furthermore, if cell-cell interaction analyses are to stay in the paper, we would need to see this further developed or justified, as both Reviewer #1 and #2 raised questions about the meaningfulness of this analysis.

Response: We designed the cell-cell gene co-expression analyses to model cell-cell interactions, based on previous studies demonstrating high degrees of agreement between the brain connectivity map and co-expression patterns of genes among brain regions. We have taken various approaches to address the questions raised by the reviewers. First, to more accurately reflect the type of analysis, we have changed the wording to "cell-cell gene co-expression analysis". Second, to demonstrate the meaningfulness of this analysis, we have now added new focused analyses on cell types that are known to interact in neural circuits in the revision. Specifically, we tested whether our cell-cell co-expression analysis can recapitulate the known hippocampal trisynaptic circuit (e.g., interactions between DG granule cells and CA pyramidal neurons). We were very encouraged to find that our gene-based analysis faithfully captured

the known mutual interactions between DG and CA3 neurons and the communication from CA3 to CA1 (new Figure 4b). This analysis supports the validity and meaningfulness of our cell-cell gene co-expression analysis. However, we agree with the Editor and Reviewers that the predictions from this analysis will require future experimental validation. As an additional step to address the reviewer concerns, we now discuss the suggestive nature of this analysis and the need for experimental testing of the predicted interactions under Results on page 11, line 219-221 and at the end of the Discussion on page 20.

E3. Finally, please address all further technical concerns, including regarding behavioral and seq analysis, raised.

Response: We have addressed all the technical concerns related to the behavioral and sequencing analysis, as detailed in the responses to the individual reviewer comments below.

Reviewers' comments:

Reviewer #1 (Remarks to the Author):

This manuscript is mostly well written, the authors use a cutting edge unbiased technique (Drop-sequencing) to determine the sequencing of thousands of cells from the hippocampus of animals that received a concussive injury. Using Drop-seq they categorize cells according to their similarities in gene expression patterns and extract cell type specific genetic markers affected by mild concussive injury. The authors were able to uncover hippocampal cell types most sensitive to the specific animal model used (concussive mild TBI (mTBI)) as well as the vulnerable genes, pathways and cell-cell interactions predictive of disease pathogenesis in a cell-type specific manner. As a proof of concept the authors targeted a specific (Ttr) pathway.

There is a lot of information that the study brings to the field with many possibilities for testing targets however the implications they draw from their findings are overstretched. The utility in their findings are helpful for understanding the possible contributions of cellular populations and how they can respond differently to the same injury (at the single cell level). Although the authors address the "flaws/downsides" of the Drop Seq method and the samples used, the continuous texts of how their work has large implications masks this information.

Response: We appreciate the reviewer's positive comments as well as the critiques about our study. We have refined our conclusions to avoid overstating the implications.

Major points:

1.1 In general the authors are drawing strong conclusions from only having analyzed correlations and this is a major problem.

Response: As pointed out by the reviewer, single cell studies generally reveal a large amount of information that opens up many new hypotheses that require further functional testing. The basic information revealed is correlative in nature, and the majority of single cell studies published to date report such correlative information, with validation experiments largely limited to confirming cell type specific markers. Accordingly, we have revised the discussion to point out possible applications of our single cell data to understand physiological processes based on current literature, and acknowledge limitations in the interpretation of our data.

However, we have included the T4 treatment experiment as an effort towards functional validation of a prioritized target obtained from the Drop-seq experiments. This, to our knowledge, is one of the first causality testing examples in single cell studies. Further, encouraged by the reviewer, we have added new multiplex ISH validation experiments to confirm changes in gene expression induced by TBI in select cell types (new Figure 7). We also added in silico validation on how our single cell analysis can recapitulate known cell-cell interaction circuits (new Figure 4b). These multiple levels of validation studies help reveal more functional information.

1.2 LN 158: How do traditional morphology-based approaches not inform on function whereas genomic features does? saying these genomic features determine function is overreaching.

Response: We agree with the reviewer that the wording was too strong. We have clarified and toned down the statement as follows on Page 9, line 155-158:

“Based on the central dogma, gene regulation is upstream of the production of proteins, which are fundamental for cell function. In contrast, morphology-based approaches may not offer the resolution to distinguish subtypes of cell populations that share similar morphology but carry certain unique functions.”

1.3 In the first section of the Results, the conclusion is that the Drop-seq approach is able to identify novel genes specific to certain cell populations which are validated by matching known gene expressions and anatomical location. Likewise, Drop-seq further differentiates populations that might seem morphologically or genetically (by previous methods) similar but now differentially cluster. There are no functional assays suggesting differences in, for example, the two CA Subtype neurons.

Response: We agree with the reviewer that the conclusion was too strong as it was only based on unique patterns of gene expression in the cell subtypes identified. We expect that this information can be used to guide separate studies to determine unique functions of cell types expressing the reported markers. These will require extensive future in-depth studies. We have revised and toned down the conclusion to the following on Page 9, line 153-162:

“These results indicate that our transcriptome-driven, unbiased Drop-seq approach has the unique ability to uncover potential new cell types, states, and markers based on genomic features that may infer function. Based on the central dogma, gene regulation is upstream of the production of proteins, which are fundamental for cell function. In contrast, morphology-based approaches may not offer the resolution to distinguish subtypes of cell populations that share similar morphology but carry certain unique functions. For instance, cells of the two CA subtype clusters uniquely express specific marker genes and may infer unique functions. However, detailed functional characterization of these potential new cell subtypes is required in future studies to test functional differences. Once confirmed, these findings provide valuable resources for future studies of the hippocampal circuitry under homeostatic and/or pathological conditions.”

In addition, we have added a general statement at the end of the Discussion section (Page 20) on the need for future in-depth functional assessments of the potential new cell types.

1.4 Fig 1: The authors mention later in the paper that TBI samples were enriched for Ependymal cells. Given that astrogliosis and microglia proliferation occurs in response to injury, how do the other cell populations compare between Sham and TBI?

Response: Compared to ependymal cells, the shift in the proportions of other detectable cell populations is not as prominent. We have calculated cell proportions in sham and TBI samples separately and found

that overall the TBI samples have increased microglial cells (10.4% in TBI vs 7.3% in sham) based on a proportion test ($p=9.0 \times 10^{-5}$), agreeing with the previously observed microglia proliferation. However, we did not observe a significant difference in astrocyte proportions, although our later gene level analysis identified transcriptome-wide shifts and numerous differentially expressed genes affected by TBI in astrocytes. Other cell types that showed significant cell proportion changes include decreases in neurons and oligodendrocytes and increases in unknown2 and endothelial cells. We have now included the cell proportion analysis in our revised manuscript on Page 9 and report the detailed results in new Supplementary Table 2. However, as many experimental factors in the Drop-seq procedure can influence the capture rate of different cell types between samples, we observed variability in cell proportion estimates between samples for the cell types with less prominent shifts. In addition, changes in the relative proportion of a cell type do not directly implicate cell proliferation or death but could be the result of shifts in other cell types. We added a cautionary note in the revised manuscript to avoid over-interpretation of these results on Page 10.

1.5 LN 161: They only used changes in cell numbers to identify vulnerability of cells but this could be a confounding. They could have supplemented by looking at genomic changes.

Response: We thank the reviewer for the suggestion. We have supplemented the cell proportion analysis with a transcriptome shift analysis (Page 10). This analysis showed that a majority of the cell types demonstrate significant global transcriptome shift. Couple with the results in a later section “Identification of genes and pathways vulnerable to mTBI in individual cell types”, which revealed significantly expressed genes (DEGs) in essentially all cell types, our various analyses suggest that most hippocampal cells demonstrate vulnerability to TBI.

1.6 LN 108: How do they know that in other regions the targets mentioned are also cell-specific?

Response: We have clarified in the main text on Page 6, line 105-106 and line 108 that the cell type specificity we refer to is for hippocampus only in the current study.

1.7 LN 139: How did they confirm the unannotated clusters are neuronal populations?

Response: In the paragraph preceding LN 139, we first categorized all cells into broad categories such as neurons, astrocytes, and microglia, based on known markers of known cell types. We then focused on the cells categorized as neurons (as they all expression typical neuronal markers) to further refine the cells into subclusters, which revealed both known neuronal subtypes and the two unannotated clusters. These unannotated clusters do express known general neuronal markers, but not specific markers that differentiate them further into any known neuronal subtypes. We added clarification to this section on Page 8, line 138-139.

1.8 Line 172-175: While oligodendrocytes, astrocytes, and microglia do appear weakly shifted in Fig1a, it is difficult to say with certainty that the claim holds true for CA1 neurons and CA subtype2 neurons. Do the authors have a more quantitative measure to support their claim? In contrast, CA Subtype1 looks to be more differentially clustered but the authors do not include the population in their list. Why?

Response: We thank the reviewer for the careful observations. We agree that the pattern for CA Subtype 1 looks more obvious and should have been included in the list. For CA1 neurons and CA Subtype 2 neurons, we previously listed these because parts of the clusters were primarily comprised of cells from sham animals (Figure 3a). As suggested, we have now added quantitative measures including cell proportion test (Page 9, new Supplementary Table 2) and Euclidian distance analysis on global

transcriptome shift (Page 10, new Supplementary Figure 10), along with differential gene expression analysis to comprehensively identify vulnerable cell types.

1.9 LN 194: it is incorrect to say that “this could be a novel property of hippocampal cells”; it could happen between cell types of other regions as well.

Response: We agree with the reviewer that this statement is misleading. We have revised the sentence to the following on Page 11, line 210-211:

“These shifts in the pattern of gene co-regulation among hippocampal cell types may implicate reorganization of the working flow in hippocampus in response to mTBI challenge.”

1.10 LN 250: If they (DEGs) are masked in bulk, can they be strong drivers? (The resolution of each technique makes this a point of discussion.)

Response: As suggested, we have added this discussion point in the revision on Page 15, line 269-275. Indeed, DEGs that are masked in the bulk analysis can still be strong drivers of diseases because these DEGs tend to be from cell populations that are less abundant yet still carry critical functions. For instance, many neuronal DEGs are not found in the bulk tissue analysis, yet neurons serve essential functions in the brain.

1.11 Figure 7A: It would be important to validate that *Ttr* is upregulated by TBI. While the authors attempt to show this through immunohistochemistry at the CA regions, the staining is not conclusive or quantified.

Response: To quantitatively validate the upregulation of *Ttr* in TBI, we have now used the RNAscope multiplex ISH to show *Ttr* changes in different cell types (new Figure 8a-j). The immunohistochemistry results now only serve as supporting evidence in Supplementary Figure 11.

1.12 Figure 7 B and C behavior results: On Day1, TBI/T4 mice learn faster than Sham, but they don't show progressive learning as shams, can they still say T4 protects learning? There are no data to show velocity of mice during Barnes maze, the shortened latency of the TBI/T4 group could be the result of higher moving speed. They need to show velocity, and number of errors to conclusively state that this is an effect in learning and memory.

Response: We thank the reviewer for the insightful comments and suggestions. We have now added number of errors and velocity plots to further examine the effect of T4 on learning. Based on the latency measure, on learning test Day 1 (which is day 7 post TBI and T4-treatment), although there is a trend for faster learning in the T4 group, there is no statistically significant difference between TBI/T4 mice and Sham mice. The number of errors analysis showed significant improvement on Day 1 in the T4 group compared to the Sham animals, but performed similarly as the Sham mice in the following days. On day 3, both T4 and Sham animals out-performed the TBI group. Overall, our data suggests that T4 treatment compensated the TBI effect on learning, and the effect is not due to moving speed as there is no difference in velocity across groups. However, as the reviewer noted, the progressive learning is less obvious in the T4 treatment group. We note that one of 7 the animals in the T4 group appeared to be an outlier showing increasing latency time instead of decreasing as in the other animals during the learning phase, contributing to the overall flatter learning curve in this group.

We do not perceive T4 as a synaptic facilitator that could act directly on learning and memory mechanisms occurring at the synapse (encoding, etc). The function of T4 as a thyroid hormone coupled with our data showing *Ttr*, the target of T4, to be altered by TBI in all cells in the hippocampus suggest

that any effect of T4 on learning or memory most likely is indirect by preserving several aspects of cell function after TBI.

1.13 Figure 7 D, G and H. The authors chose to analyze overlapping genes between TBI vs Sham and TBI/T4vsTBI, if TBI causes metabolism abnormalities, then they treat mice with a metabolism simulating reagent, of course the enrichment analysis will show mainly metabolism related GO terms. This result is heavily biased by their analysis protocol. It would be more informative to compare all altered genes in T4vsTBI.

Response: Our analysis was designed to address which TBI pathways were reversed by T4 treatment, which requires that we focus on the genes that showed reversal in expression patterns. This analysis does not involve pre-selecting metabolism related genes. Instead, we compare all genes affected by TBI with those affected by T4 to look for those that show opposite changes. The annotation of the overlapping genes was also not biased towards metabolism genes. Therefore, there is no bias in our analytical protocol towards metabolism genes. We consider the enrichment of the metabolism pathways as a confirmation of our hypothesis that normalizing metabolism by T4 is underlying the observed counteracting effects of T4 against TBI.

To provide a more comprehensive understanding of the T4 effect without selecting the overlapping genes with TBI, as suggested by the reviewer, we have now added pathway analysis on all DEGs from the TBI and T4 treatment and report the results in the main text on Page 18, in the updated Figure 8, and a new Supplementary Table 6. Although we hypothesize that metabolism is a potential major driver of TBI pathology, we do not expect that it is the only driver. Indeed, there are unique pathways affected by TBI that were not corrected by T4, and pathways that were specific to T4 only. The main pathways overlapping between the two treatments are metabolic processes.

1.14 Cell-cell interaction section: This section is difficult to understand for its implications. The gene co-expression method does give some insight to how genes of one cell type correlates with genes of another. However, the analysis does not provide direct information about how cell-cell interactions are disrupted. At best, the methodology here can conclude a shift in cell-cell interaction in the injured brain, but cannot actually inform which cell communication pathways are disrupted. To address this at a minimum the authors need to change the title so it does not imply a downshift in cell-cell interaction. This is especially true since the authors state both increase in gene co-expressions (astrocyte and ependymal to neurons, microglia to oligodendrocytes) as well as decreases (microglia to neurons, oligodendrocytes to neurons).

Response: Our gene co-expression analysis was an attempt to leverage the single cell data to infer potential cell-cell interactions, based on previous observations that gene co-expression patterns infer brain connectivity map. We agree with the reviewer that the gene co-expression analysis presented only yields suggestive information on potential alterations of cell-cell interactions. We have changed the title as suggested (page 10, line 194), and discussed the limitations of this *in silico* analysis. In addition, we have added focused analysis on known cell-cell interaction circuits (see detailed response to Editor comment E2) to demonstrate that our gene co-expression analysis can indeed capture known interactions, thereby enhancing the interpretability of the results.

1.15 Given the authors' findings about the ability to detect cell-type-specific genes for treatment, what was the rationale for choosing Ttr which the authors acknowledge is a pan-hippocampal target? This is especially notable given that the authors specifically highlight the effect of TBI on DG Granule cell gene expressions (Fig 1a and Fig3a). Furthermore, does targeting Ttr with T4 treatment effect certain cell types more or is it effective across all cell types? The data only looks at pan-hippocampal sequencing data.

Response: We agree with the reviewer that targeting cell type specific genes is a highly plausible path to test specific therapeutic effects mediated by individual cell types, and we acknowledge this important direction on page 16, lines 313-316. We are in the process of planning studies to manipulate and test the various predicted cell type specific genes. However, targeting specific genes in specific brain cell populations are highly technically challenging and time-consuming, and we believe timely publication of the current study is important for the scientific community. As such, we respectfully request to keep such experiments as a future direction.

In the current study, we choose to leverage the full information across all cell types examined to prioritize *Ttr* as the primary target for testing. The main rationale is that we believe the strongest and broadest therapeutic effects need to harmonize with the broad aspects of the TBI pathology. Therefore, targeting the most consistent genes across cell types is a sound therapeutic approach for TBI. We consider this as an innovative and productive use of the single cell data. *Ttr* shows great variability in the basal expression levels across cell types (Figure 6b, Figure 8). When averaged across all cell types, it is not the most prominent gene affected by TBI, thus explaining why *Ttr* has not been ranked as a top candidate by previous TBI studies that were based on bulk tissue analyses. Our single cell study, for the first time, uniquely provides the resolution to unveil that *Ttr* is a transporter of T4 that is expressed at variable levels between cell types but is consistently altered by TBI across cell types, suggesting that T4 treatment will have broad effects across hippocampal cell types. As the T4 experiments primarily serves as a proof-of-concept of using drop-seq data to prioritize drug targets, we consider that the phenotypic and pan-hippocampal sequencing data from the T4 experiments provide sufficient support for the predicted effects and the potential general mechanisms. We plan to conduct drop-seq experiments in the future to examine effects of T4 treatment in individual cell types.

In the revision, we further clarify the rationale for focusing on *Ttr*, discuss the limitations of the T4 experiments, and point to future experiments to address the limitations on pages 17-18.

1.16 Methods:

RNA-seq analysis of T4 treatment experiments: Authors should write whether the tissues collected for RNA-seq were from the mice that ran the behavioral experiment. The don't line up (n=6/group for behavior, n=4/group for RNA-seq). Furthermore, what time point were the animals taken down for RNA-seq (if after behavior, it would be at 11 days post-injury and treatment; if sooner, how long after the last injection of T4)?

Response: We have added more experimental details in the revision to clarify the samples and time points (Page 32). The RNAseq samples were a subset of mice that underwent behavior tests. For behavior examination, we used n=6 per group. For RNAseq, we selected 4 samples out of the 6 animals/group. This is a typical sample size for RNAseq studies in control/treatment experiments in animal models. The time point for RNAseq is 11 days post-injury/treatment. We recognize that additional time points (24hr, 3 days, 7 days) are needed to fully examine the T4 effects and we plan to conduct drop-seq analysis to investigate the effects of T4 on individual cell types at multiple time points as a future direction.

1.17 Minor points:

a. Line 57: what "spectrum of the hippocampus" refers to in line 57.

Response: We have changed the phrase to "*psychiatric disorders associated with the hippocampus*" on page 4, line 53.

b. LN 131: Example of a "reachy statement."

Response: We have tuned down the statement on Page 9.

c. Line 153-155: Has damage to subiculum been explicitly linked to TBI models?

Response: Damage to the subiculum is often unreported in TBI models. Our concussion model does not involve cell death or axonal degeneration in the hippocampus; however, we have reported sudden changes in synaptic plasticity. Subicular damage is getting more attention based on studies showing histopathological damage in subiculum of postmortem individuals suffering from chronic traumatic encephalopathy or Alzheimer's disease.

d. LN 169: Confusing text.

Response: We have revised the text to the following on Page 10, lines 183-184:

“In particular, mTBI had such a profound effect on the transcriptome of DG granule cells, that they became clearly separated into two distinct clusters: 94% of cells in one cluster are from the Sham animals and 86% of cells in the other cluster are from the mTBI samples (Fig. 1a, Fig. 3a).”

e. LN 212-216: Not new information to the field.

Response: We agree with the reviewer that these are well known pathways in TBI. The purpose of this section is to confirm that our study recapitulated established pathways and, more importantly, points to the cell origins and provides new insights. We have clarified this on Page 12, lines 236-239:

“Nevertheless, our data uniquely points to the specific cell types engaging these pathways and offers novel insights into the functions of individual cell types, including previously understudied cell populations, in mTBI pathogenesis.”

f. LN 254: what is that value? Don't state it clearly.

Response: We have revised the text to state the added value from single cell analysis on Page 14, lines 269-277:

“Importantly, >50% of the DEGs identified at the single cell level were masked in bulk tissue analysis (Fig. 5f; bulk tissue-level DEGs in Supplementary Table 4) and these unique cell-level DEGs were primarily from cell types of low abundance such as neurons. On the other hand, the common DEGs between single-cell and tissue-level analyses were mainly from abundant cell types such as astrocytes and oligodendrocytes. As less abundant cells such as neurons carry essential functions in the brain, the cell type-specific DEGs can be strong drivers of disease but will be missed in the bulk tissue analysis. Therefore, genomic information in individual cell types has the advantage to extract hidden mechanisms involved in TBI pathology that otherwise would be masked in bulk tissue studies.”

g. Line 276: Please provide a reference for the line referring to a link between mTBI and increased tendency for neuroticism post-TBI.

Response: We apologize for missing the reference, and have added the reference for the link between mTBI and increased tendency for neuroticism post-TBI.

h. Figure 6: needs a figure legend (color).

Response: A figure legend (with disease condition colors) has been added to Figure 6.

i. Figure 7E and F: error bars for these plots should be shown.

Response: We have added error bars to these plots.

j. Line 374 – 378: The sentence is difficult to parse. Rewrite it for clarity.

Response: The sentence has been rewritten as follows on Page 19-20:

“The transcriptome perturbations by mTBI in individual cell types help pinpoint the cell origins of processes that likely guide mTBI pathogenesis, such as metabolic dysfunction in astrocytes and neurons and amyloid deposition involving ependymal and endothelial cells. These cell-level transcriptome patterns depict gene programs that may regulate and predict susceptibility to post-mTBI neurological disorders such as AD, PD, PTSD, neuroticism, and epilepsy.”

Reviewer #2 (Remarks to the Author):

In this study, Yang and colleagues investigated the cell types and molecular pathways in the hippocampus that were altered by concussive brain injury by applying single-cell RNA sequencing to a mouse model. They compared the single-cell expression profiles of hippocampus between three mTBI and three control animals. Identify the vulnerable cell types, and specific pathways that correlated with the injury. They further prioritized potential molecular targets for therapeutic interventions based on differentially expression genes, and experimentally demonstrated that the Ttr was highly expressed post injury, potentially as a compensatory mechanism for thyroxine T4, and the in junction of T4 post-mTBI can reversed the transcriptional changes induced by mTBI.

Overall, this is a timely study of the molecular mechanisms underlying mTBI using a cutting-edge single-cell transcriptional profiling approach. It revealed novel insights related to the cell types and pathways specific to mTBI. The findings provide new targets for therapeutic intervention.

2.1 One major criticism of this study is a general lack of experimental validation on the findings by single-cell transcriptional profiling. The authors did presente many ISH images, including many in Figure 2. However, these were the existing single-gene low-resolution ISH data produced by the Allen Brain Institute, on normal mouse brain. Multiplex RNA in situ data on the mTBI and sham animal are necessary to validate some of the key findings in this study.

Response: We appreciate the reviewer’s comments and have added additional multiplex RNA in situ data on both mTBI and Sham animals to support the key genes altered by mTBI (new Figures 7 and 8).

2.2 Second, some of the analyses were a bit loose. For example, the analysis of cell-cell interaction in Page 10 is really interesting, but how robust is the method used for the analysis? Cell-cell interactions can't be studied without the context of spatial organization. An observation that cell A is secreting some peptides and cell B has the surface receptors of such peptides does not necessarily mean real interaction unless there is evidence that the peptides can reach cell B within the 3D space of hippocampus.

Response: We appreciate this highly insightful comment. Our gene co-expression analysis was an attempt to model the single cell data to infer potential cell-cell interactions, based on previous observations that gene co-expression patterns capture brain connectivity map. We agree with the reviewer that this analysis only yields suggestive information on potential alterations of cell-cell interactions. Sophisticated circuit tracing techniques are required to provide the necessary spatial information, and gene perturbation experiments in one cell type coupled with response measures in the predicted target cell type are needed to validate the causal interactions. These are the research directions we are actively pursuing but are technically challenging and beyond the scope of the current study.

To strengthen the validity and the interpretability of our analysis, in the revision, we have added focused analysis of known cell-cell interaction circuits to demonstrate that our gene co-expression analysis can indeed capture known interactions. In addition, we have changed the title of the analysis to “cell-cell gene co-expression analysis”, discussed the suggestive nature of this *in silico* analysis, and the need for

experimental testing of potential cell-cell interactions inferred by the analysis.

Reviewer #3 (Remarks to the Author):

This paper by Arneson and colleagues reports the results of a high throughput single cell sequencing study performed on thousands of individual hippocampal cells isolated from the mouse brain after concussive mild TBI. The study also identifies the transthyretin (Ttr) gene, a gene coding for a transport protein that carries the T4 thyroid hormone across the blood-brain barrier, and validates T4 as a novel treatment for mTBI. Overall, the manuscript is well-written and the data are novel, convincing and of potential high impact to the neuroscience field. The experimental flow of the paper follows a straightforward and logical path to acquire some interesting findings. However I do have some relatively minor issues that should be taken into consideration in a revised manuscript.

3.1 The authors list in the introduction a number of critical longstanding questions that they claim their study will help resolve. One of these questions regards the identity of the hippocampal cell types that are the most vulnerable to concussive mild TBI. However, based on data presented in Fig. 1 and Table 1, it can be assumed that basically all of the hippocampal cell types are vulnerable to mTBI. Were some of the differentially-expressed genes enriched for pathways involved in cell death? Along the same line, the abstract should be more informative on this matter.

Response: We appreciate these great suggestions. In the revision, we have added in the discussion that our data indicates that the majority of hippocampal cell types demonstrate various degrees of sensitivity to mTBI (Page 10, lines 188-190; Page 19, lines 384-387) and have revised the abstract accordingly (Page 3). Indeed, cell death was one of the pathways affected by mTBI. Specifically, we see enrichment of cell death pathways in astrocytes, microglia, and various neuronal populations. We have added these results in the revised manuscript on Page 12, lines 235-236.

3.2 Another goal of the study was to identify diagnostic and prognostic biomarkers for mTBI. Among the biomarkers that were identified by the unbiased single-cell sequencing (Drop-Seq) method are Id2, p2ry12, apoe, and Itm2a. What is unclear, however, is whether any of these biomarkers can be measured other than by post-mortem brain examination. Adding a confirmation of upregulation of protein expression for one of these biomarkers in the cerebrospinal fluid or blood of mTBI mice would be a significant plus value to the study.

Response: We agree with the reviewer that our study identifies numerous genes altered by mTBI in individual cell types and some of which may serve as viable diagnostic markers. However, these gene expression changes are mostly within individual cell types, and which of these are readily secreted and captured as protein level changes in the cerebrospinal fluid or blood are highly uncertain. If we test a handful of candidate genes out of the hundreds of DEGs identified from the drop-seq data, the chance of finding a viable protein biomarker can be unpredictable and inconclusive. We feel that a systematic screen via proteomic approaches at multiple time points in both cerebrospinal fluid and blood will be a more appropriate path in a future, focused biomarker study. We point out the need for such study at the end of the Discussion on Page 20.

3.3 The circo plots included in Fig. 4b-c depict genes coding for secreted proteins that have the potential to interact with receptor-encoding genes in target cells during mTBI pathogenesis. This dataset is critical to the overall importance of the work, and also for future studies that could derive from it, but should be improved in a number of ways: 1) the lines are just too hard to follow from one cell type to another, 2) each secreted factor should be linked to an identifiable specific receptor

in the target cell, and the name of that receptor should appear in the revised figure.

Response: We would like to clarify that the two circos plots are intended to model the global rewiring in the overall cell-cell gene-co-expression patterns between TBI and Sham groups rather than specific connections. In addition, on the target cell end, the analysis was not limited to only the receptor genes, as we believe that a secreted factor, if indeed transmitting information between cells, is likely to trigger changes in a large number of downstream effector genes than simply its receptor in a target cell. Therefore, our analysis captures many genes in the target cells that show strong correlation with each secreted protein from the source cell. For this reason, we were not able to label all genes on the target cell side, but we have provided a new Supplemental Table 3 to show all the correlation pairs to guide future follow-up studies.

In light of the reviewer's suggestions, we have added additional focused analysis on a well-annotated hippocampal trisynaptic circuit involving specific neurotransmitters (new Figure 4b-d). This targeted analysis helped to confirm the ability of our approach to capture true cell-cell communications.

3.4 Figure 7a: The immunofluorescence staining for Ttr is not convincing (weak signal and some images are out of focus). Also, Fig. 7 and Supplementary Fig. 10 are somewhat redundant.

Response: We appreciate the feedback. We have replaced the immunofluorescence staining with multiplex RNA ISH assays (new Figure 8).

3.5 Figure 7b: Did the T4 treatment correct TBI-induced learning deficit or not? Significance is not shown between the TBI/Veh and TBI/T4 groups in the graph.

Response: As the reviewer noted, by the latency measure there was no difference between TBI/Veh and TBI/T4 during the learning phase. However, when the number of errors was used as suggested by Reviewer #1 (new Figure 8l, 8n), there were significant differences at day 1 and day 3 between TBI/Veh and TBI/T4 during the learning phase. Therefore, there is evidence supporting an improvement of learning by T4. Similarly, T4 appears to improve memory in the analyses using both the latency time and the number of errors. Our molecular data suggests that T4 treatment preserves cellular processes such as metabolism that are accessory to learning and memory, rather than directly regulating neuronal signaling. This possibility is also in agreement with the Drop-seq data showing compensatory changes in Ttr expression across several cell types after TBI and the known role of T4 and Ttr in metabolism.

Reviewers' comments:

Reviewer #1 (Remarks to the Author):

The authors do remarkable job trying to address the reviewer concerns and it is clear they are experts in analyzing sequencing data. However, they do not convincingly convert their sequencing findings to actual functional validation of a single target identified by Drop-seq. Rather, they show that Drop-seq may aid in identifying pathways affected by mTBI which in turn can elucidate novel treatment approaches. While still important, they essentially are overstating their approach and the significance of their single-target experiments (figure 8). Most importantly the additional results with in situ hybridization are very weak.

-The in situ figure is weak and does not show any significant colocalization. In fact, many of the encircled regions don't show colocalization at all. Furthermore, the authors do not quantify colocalization in any meaningful way, making it difficult to tell if any of the sparse possible-overlap is random noise or biologically-relevant colocalization.

-Additionally, to actually state that the in situ experiments agree with the Drop-seq results (alterations of select DEGs by mTBI) requires quantification of the staining rather than simply a qualitative image.

Fig 8

-8a-j: While the staining shows the presence of Ttr in multiple hippocampal regions, the data does not show expression across cell types (as the authors suggest).

-The authors need to better explain the behavioral assay and their analysis of the data for 8k-n (we appreciate the authors verifying that velocity is not the causal factor to improvement in behavior in the treatment group).

-Fig 8o does not suggest that T4 is preferentially engaging the main T4 transporter Ttr. The authors do not discuss other potential pathways of T4 entry into the hippocampus (which supposedly is responsible for the improvement of cognition).

-It is important to note that treatment with T4 does NOT directly target Ttr, which is the justification the authors originally make (i.e. that their treatment is modulating Ttr which would directly affect mTBI outcome). In fact, the results would suggest that mTBI affects T4 presence in the brain, and Ttr upregulation is a downstream effect of the deficit in T4. It would be more appropriate (if we keep T4 as the treatment) to show the presence of T4 in the hippocampus in sham animals and after injury as well as changes peripherally (since the injection is done via i.p.). As such, while the author's use of their sequencing data did lead to a novel treatment, it identified an impaired pathway rather than a specific target gene or molecule directly responsible for the cause of TBI-induced cognitive deficits. In fact, the supposed strength of the sequencing data to identify specific targets to modulate and improve mTBI outcome is not supported by the experimental approach underlying Figure 8.

Reviewer #2 (Remarks to the Author):

With the new data and a more careful interpretation/presentation of their data, this revision is significantly improved. I have no more concern, and recommend this manuscript to be accepted for publication.

Reviewer #3 (Remarks to the Author):

Arneson and colleagues have taken seriously the comments made by the three reviewers, and I commend them for that. However, as it stands, the revised paper is still missing the mark when it comes to providing a follow up validation of the importance of single cell gene expression data for i) the understanding of the pathophysiological role of certain genes in TBI, and/or ii) the design of future therapies. Also, the validation of gene or protein expression at the single cell level in tissue sections remains problematic. ISH (Figs. 7-8) and IHC (Suppl. Fig. 11) data are still of insufficient quality for publication in Nat Commun. My considered opinion therefore remains that the results included in the manuscript are of potentially high impact to the neuroscience community and deserve to be published in Nature Communications, pending the ability of the authors to produce the data requested on the two main issues described above.

Response to Editor/Reviewer Comments

We are pleased that many of the responses we provided in the first revision were satisfactory to the reviewers, and we thank the Editor and the Reviewers for the additional constructive and thoughtful comments to help us further improve our study. We now have revised the manuscript to address each of the remaining concerns, as detailed below in our point-by-point response. Editor/Reviewer comments are in bold.

Editor's comments:

E1. We insist that you edit language to acknowledge other possible targets of T4 aside from *Ttr* and to tone down language suggesting direct therapeutic application (e.g. "revealing" pathogenesis).

Response: We appreciate the Editor's suggestions. We have edited the language to acknowledge other possible targets of T4 and toned down our language regarding therapeutic application as follows on Page 19, line 384-394:

“Our analysis of known T4 transporters indicates that T4 treatment primarily downregulates *Ttr* and has weaker effects on genes encoding other transporters (**Fig. 9e**). Genes encoding thyroid hormone receptors that are downstream of T4 also had less consistent changes across cell types compared to *Ttr* (**Supplementary Table 6**). These results agree with our hypothesis that the upregulation of *Ttr* seen in TBI is an indicator of thyroid hormone deficiency in the brain and T4 treatment reverses this change. We acknowledge that although our gene expression data suggests that *Ttr* is more strongly modulated by T4 compared to the other known T4 transporters and receptors, substrate binding experiments are needed to test whether *Ttr* is the main T4 transporter. Through whole transcriptome profiling, we also identified a cascade of genes and pathways potentially involved in T4 effects.”

We have also edited the title to tone down language suggesting direct therapeutic application:

“Single Cell Molecular Alterations Reveal Target Cells and Pathways of Concussive Brain Injury”

E2. Also, it will be crucial that you can provide new ISH images that are satisfactory to the referees, as this was a point raised by two of the referees.

Response: We have taken major efforts to improve the image quality of the ISH experiments and to provide a thorough quantitative account of gene expression changes in single cells.

Please note that we are using RNAscope technology, which is a state-of-the-art high-resolution quantitative ISH that is different from traditional low resolution ISH that rely on colorimetric non-linear quantification. We chose RNAscope because it has the capacity to accurately measure single mRNA molecules. In this revision, we provide a better explanation of the underlying technology and quantification methodology. We feel that some concerns over the co-localization between cell marker genes and DEGs could be a result of confusions about how to define co-localization using this new high-resolution quantitative technology. Specifically, with traditional fluorescent ISH, the analysis is based on color overlaps between a marker gene and a DEG within a cell. In contrast, RNAscope detects single mRNA molecules within each cell, and co-localization is defined by the simultaneous presence of sufficient numbers of the two fluorophores representing a marker gene and a DEG in the same segmented cell. Quantitatively, RNAscope relies on fluorescent density which is proportional to mRNA abundance. Accordingly, we have implemented state-of-the-art automated quantification procedures to assess the number of fluorescent spots per cell as the quantitative measure for mRNA abundance. We have added

these clarifications to help data interpretation on Page 15-16, line 310-329 under Results and Page 31-32, line 667-688 under Methods.

Furthermore, we have increased our sample size for RNAscope to n=8 per group in this second resubmission, taking over 900 multi-channel RNAscope z-stack ISH images and quantified multiple DEGs using the established imaging processing and quantification software FISH-quant to validate our cell-type specific single cell Drop-seq results. We have also provided improved RNAscope ISH images (high magnification images in main text **Fig. 7, Fig. 8**; low magnification images in **Supplementary Fig. 11, and Supplementary Fig. 12**) which more clearly demonstrate the colocalization of the cell-type specific marker genes and DEGs within the same cell segmentation as determined by the established imaging software CellProfiler. We are encouraged that our quantitative analysis of RNAscope ISH validated significant changes in all but one cell-specific DEGs tested (new **Fig. 7**) as well as significant *Ttr* increases in all of the tested cell types (new **Fig. 8**). We hope these new additions are now satisfactory to the Editor and the Reviewers.

E3. Please highlight all changes in the manuscript text file.

Response: We have highlighted all changes in the manuscript text file.

Reviewers' comments:

Reviewer #1 (Remarks to the Author):

The authors do remarkable job trying to address the reviewer concerns and it is clear they are experts in analyzing sequencing data. However, they do not convincingly convert their sequencing findings to actual functional validation of a single target identified by Drop-seq. Rather, they show that Drop-seq may aid in identifying pathways affected by mTBI which in turn can elucidate novel treatment approaches. While still important, they essentially are overstating their approach and the significance of their single-target experiments (figure 8). Most importantly the additional results with in situ hybridization are very weak.

Response: We appreciate the reviewer's positive comments as well as the critiques. As suggested, we have refined our conclusions to avoid overstating the approach and the significance of single-target experiments. We have also conducted additional ISH experiments and quantitative analysis to more convincingly demonstrate the validation of cell-type specific gene expression changes. We now provide new ISH figures in main text **Fig. 7 and Fig. 8** (high magnification) and **Supplementary Fig. 11 and Fig. 12** (low magnification), and have updated the corresponding texts, as detailed below.

Major points:

1.1 The in situ figure is weak and does not show any significant colocalization. In fact, many of the encircled regions don't show colocalization at all.

Response: We would like to clarify that we used RNAscope, a state-of-the-art high-resolution quantitative ISH, to define and quantify cell type specific expression and alterations in candidate DEGs. As this is a high-resolution quantitative approach, each fluorescent spot represents a single mRNA molecule, so we do not rely on overlapping colors/probes to determine colocalization as would be needed in low resolution traditional FISH (also see details in response to Editor comment E2 above). Using RNAscope, colocalization is defined as the presence of probes for both cell type-specific marker gene and DEG

within the same cell boundary. In the revised manuscript, we have provided details regarding how we define cell boundaries and how we colocalize and quantify mRNAs within the same cell. We also provided better images demonstrating the entire process in new figures **Fig. 7**, **Fig. 8** (high magnification), **Supplementary Fig. 11**, and **Supplementary Fig. 12** (low magnification). The details of the method are updated on Page 31-32, line 667-688. We hope these new additions help clarify that the images indeed demonstrate colocalization between a cell marker gene and a candidate DEG.

1.2 Furthermore, the authors do not quantify colocalization in any meaningful way, making it difficult to tell if any of the sparse possible-overlap is random noise or biologically-relevant colocalization.

Response: As explained above, in contrast to traditional ISH, RNAscope determines colocalization by examining whether multiple mRNA molecules from a candidate DEG are present in the same cell as the mRNA molecules of a cell type marker gene. We require a minimum number of mRNA molecules from each gene to be present within a cell as determined by an established imaging software FISH-quant to help reduce random noise.

1.3 Additionally, to actually state that the in situ experiments agree with the Drop-seq results (alterations of select DEGs by mTBI) requires quantification of the staining rather than simply a qualitative image.

Response: We appreciate the suggestion and have now added quantification of cell-type specific gene expression using an imaging software FISH-quant, as presented in new figures, **Fig. 7**, **Fig. 8**, **Supplementary Fig. 11**, **Supplementary Fig. 12**. We are encouraged that this objective quantification process helped confirm the significant differential expression of all but one of the cell-type specific DEGs tested (**Fig. 7**, **Supplementary Fig. 11**) as well as significant increases in *Ttr* in all of the tested cell types (**Fig. 8**, **Supplementary Fig. 12**), thereby quantitatively supporting the accuracy of our Drop-seq results.

We chose to use RNAscope due to its superior specificity and highly quantitative nature compared to other RNA ISH technologies. Nevertheless, we would like to emphasize that it is non-trivial to quantify RNAscope ISH data due to the lack of existing mature quantification methods. In fact, through our literature search, we did not find any published examples to quantitatively measure mRNA molecules of a gene in a cell type specific manner using RNAscope. Most examples of multi-channel RNAscope primarily qualitatively assess whether the two genes are co-localized, or examine percentages of co-localization (which can be misleading as the individual mRNA molecules of a cell marker gene do not have to be located at the same locations as the single mRNA molecules of a DEG in the same cell). Indeed, the task of quantitatively measuring the expression of a gene at single cell resolution in a specific cell type is technically challenging. To best address the reviewer's request, we benchmarked various analytical approaches ranging from manual cell segmentation and mRNA counting, semi-automated analysis, to a fully automated approach using CellProfiler for cell segmentation and FISH-quant for quantification. Thorough comparison of the various approaches revealed that the automated approach incorporating CellProfiler and FISH-quant is the most feasible, objective, and robust method to systematically quantify over 900 images, as detailed below.

One of the key technical challenges in quantifying cell type specific DEGs is the ability to accurately segment cells which is necessary for the identification of colocalization of cell marker genes and DEGs. This step is non-trivial in tissue samples, especially in regions of high cell density like the dentate gyrus. To achieve this goal, we used the highly cited, state-of-the-art cell segmentation tool CellProfiler. CellProfiler takes maximal 2D projections of DAPI and fluorescent probe z-stack images as input, and outputs the cell boundaries. CellProfiler leverages information from the DAPI stain to determine the boundary of the cell nucleus and the fluorescent spots representing the expressed genes to infer the extent

of a cell's cytoplasm. To determine the nuclei boundaries, a threshold is determined using the maximum correlation threshold method on the whole image, which has been smoothed with a Gaussian function with clumped nuclei decoupled using peak intensities. To leverage the fluorescent probe information to infer cell boundaries, the Watershed-Image method is employed. Briefly, fluorescent spots are assigned to nuclei inferred from the DAPI images, which serve as seeds for a "watershed". To determine the boundaries between cells, the areas of lowest intensity in the inverted images are used. Subsequently, the boundaries of each cell for each of the 900+ images are fed back into FISH-quant which can leverage the full 3D information from the z-stack images to distinguish true fluorescent spots coming from mRNA molecules from background fluorescence/hybridization representing noise. Briefly, the images are first blurred using a Gaussian filter with a large kernel to approximate the background noise which is subtracted from the original image. This image is again filtered with a small Gaussian kernel to boost the signal-to-noise ratio. A 3D Gaussian is fit to each fluorescent spot within specified intensity constraints, and a quality score for each spot is then determined by examining the standard deviation of the pixel intensities around each spot which can be used to filter out spurious signals. After these pre-detection settings are determined, a new 3D Gaussian is fit to each detected spot on the raw image because processing on the filtered image yields suboptimal results. Fitting parameters estimated for each fluorescent spot are returned and can be used to exclude spots which are false-positives and background. These parameters include the sizes of the fluorescent spot in the XY-plane and in the Z-plane and the amplitude of the 3D Gaussian fit to the spot. While the number of images increases, the tuning of these parameters becomes more robust as we expect the distributions of these parameters to fit a skewed Gaussian, thereby allowing for the detection of outliers.

1.4 Fig. 8a-j: While the staining shows the presence of Ttr in multiple hippocampal regions, the data does not show expression across cell types (as the authors suggest).

Response: We agree with the reviewer that it is necessary to show the alteration of *Ttr* in specific cell types. We have added images and quantification of *Ttr* in the following cell types: ependymal cells, microglia, CA1 pyramidal neurons, and CA subtype 2 cells (in the CA1, CA2, and CA3 regions). The results are presented in new figures **Fig. 8** (high magnification) and **Supplementary Fig. 12** (low magnification). We were able to quantitatively confirm the significant increases in the expression of *Ttr* in all of the cell types tested.

1.5 The authors need to better explain the behavioral assay and their analysis of the data for 8k-n (we appreciate the authors verifying that velocity is not the causal factor to improvement in behavior in the treatment group).

Response: We appreciate this feedback and have now extended our descriptions of the behavioral assays and data analysis processes involved in 8k-n (now new **Fig. 9a-d**) as follows on Page 18, line 370-378:

"Acute intraperitoneal injection of T4 within the first 6 hours post-mTBI protected learning and memory, as determined by the Barnes Maze test one-week post-mTBI. Briefly, for the learning component, animals were trained with two trials per day for four consecutive days, and memory retention was assessed two days after the last learning trial. Differences in learning and memory between groups were determined using a two-tailed Student's t-test. The learning effects were evidenced by a sustained reduction in the latency to find the escape hole across all time points in the T4 group compared to TBI mice without T4 (**Fig. 9a-b**). The effects of T4 on memory were demonstrated by the recovery of the latency time to a level comparable to the Sham group (**Fig. 9c-d**)."

1.6 Fig. 8o does not suggest that T4 is preferentially engaging the main T4 transporter Ttr. The authors do not discuss other potential pathways of T4 entry into the hippocampus (which supposedly is responsible for the improvement of cognition).

Response: We appreciate this insightful comment from the reviewer and have now refined our conclusion on *Ttr*. In **Fig. 8o** (now **Fig. 9e**), we used gene expression changes in all known T4 transporters in the brain, based on our knowledge, as a proxy to infer which transporter is more responsible for T4 actions. We found that TBI induced high expression of *Ttr*, which was reversed by T4 treatment to the same level as in sham animals. In contrast, the other known T4 transporters did not show this pattern. We further explored gene expression patterns of thyroid hormone receptors that are downstream of T4 and also found less consistent changes (new **Supplementary Table 6**). We have now edited the language to acknowledge that although our gene expression data suggests that *Ttr* is more strongly modulated by TBI and T4 treatments compared to the other known T4 transporters and thyroid hormone receptors, substrate binding experiments are needed to test whether *Ttr* is the main T4 transporter. Our full transcriptome analysis indeed supports that T4 treatment influences many other pathways that may be responsible for the improvement of behavior. The revised text is as follows on Page 19, line 384-394:

“Our analysis of known T4 transporters indicates that T4 treatment primarily downregulates *Ttr* and has weaker effects on genes encoding other transporters (**Fig. 9e**). Genes encoding thyroid hormone receptors that are downstream of T4 also had less consistent changes across cell types compared to *Ttr* (**Supplementary Table 6**). These results agree with our hypothesis that the upregulation of *Ttr* seen in TBI is an indicator of thyroid hormone deficiency in the brain and T4 treatment reverses this change. We acknowledge that although our gene expression data suggests that *Ttr* is more strongly modulated by T4 compared to the other known T4 transporters and receptors, substrate binding experiments are needed to test whether *Ttr* is the main T4 transporter. Through whole transcriptome profiling, we also identified a cascade of genes and pathways potentially involved in T4 effects.”

1.7 It is important to note that treatment with T4 does NOT directly target *Ttr*, which is the justification the authors originally make (i.e. that their treatment is modulating *Ttr* which would directly affect mTBI outcome). In fact, the results would suggest that mTBI affects T4 presence in the brain, and *Ttr* upregulation is a downstream effect of the deficit in T4. It would be more appropriate (if we keep T4 as the treatment) to show the presence of T4 in the hippocampus in sham animals and after injury as well as changes peripherally (since the injection is done via i.p.). As such, while the author's use of their sequencing data did lead to a novel treatment, it identified an impaired pathway rather than a specific target gene or molecule directly responsible for the cause of TBI-induced cognitive deficits. In fact, the supposed strength of the sequencing data to identify specific targets to modulate and improve mTBI outcome is not supported by the experimental approach underlying Figure 8.

Response: We thank the reviewer for these insightful comments. Accordingly, we have refined our hypothesis and interpretation of the data. Indeed, we chose T4 as a potential therapy based on the hypothesis that *Ttr* upregulation across hippocampal cells informs on T4 deficiency post-TBI (Page 17-18, line 358-362). Given that the receptors for thyroid hormone are in the cell nucleus, it is likely that thyroid hormone function is heavily dependent on its transporters for the internalization of thyroid hormone into cells. Our data showing that TBI upregulates the *Ttr* gene across main cell types affected by TBI suggest that *Ttr* plays an important role in the internalization of thyroid hormones into brain cells. Accordingly, we hypothesized that supplying T4 will likely be beneficial for TBI. We have toned down the wording regarding *Ttr* being a direct TBI/T4 target, and state that our single cell data helped us identify important impaired pathways, as suggested by the reviewer.

Reviewer #2 (Remarks to the Author):

With the new data and a more careful interpretation/presentation of their data, this revision is significantly improved. I have no more concern, and recommend this manuscript to be accepted for publication.

Response: We appreciate the reviewer's favorable consideration of our revision.

Reviewer #3 (Remarks to the Author):

Arneson and colleagues have taken seriously the comments made by the three reviewers, and I commend them for that.

My considered opinion therefore remains that the results included in the manuscript are of potentially high impact to the neuroscience community and deserve to be published in Nature Communications, pending the ability of the authors to produce the data requested on the two main issues described above.

Response: We appreciate the reviewer's positive comments.

Major points:

3.1 However, as it stands, the revised paper is still missing the mark when it comes to providing a follow up validation of the importance of single cell gene expression data for i) the understanding of the pathophysiological role of certain genes in TBI, and/or ii) the design of future therapies.

Response: We completely agree with the reviewer that validating the importance of the genes from our single cell studies is important and we are indeed actively pursuing such experiments using chemogenetic approaches that can target specific genes in specific cell types. However, such follow-up experiments are extremely challenging and tedious to complete in a reasonable time frame. Similar to most single-cell studies published to date, which primarily report novel insights with validation experiments mainly restricted to ISH rather than functional validation, our study unravels numerous new findings and hypotheses. We feel it is important to publish our current findings in a timely manner so as to open the opportunity for the neuroscience community to collectively test the functional and pathophysiological implications of the genes and pathways uncovered. In this new revision, we provide stronger ISH validation data and refine our wording regarding the importance of single cell studies in informing the perturbed cell types and pathways that can facilitate future functional studies and the design of therapies.

3.2 Also, the validation of gene or protein expression at the single cell level in tissue sections remains problematic. ISH (Figs. 7-8) and IHC (Suppl. Fig. 11) data are still of insufficient quality for publication in Nat Commun.

Response: We recognize the need to provide more convincing ISH data. As detailed in our responses to Editor comment E2 and Reviewer 1 comment 1.3, we dedicated considerable efforts to conduct RNAscope quantitative ISH for multiple DEGs across 16 animals (n=8/group, sham vs. TBI), culminating in over 900 sets of DAPI, GFP and CY3 z-stack images (new **Fig. 7, Fig. 8, Supplementary Fig. 11, and Supplementary Fig. 12**). Our quantitative analysis of these RNAscope ISH images validated the differential expression of all but one cell type-specific DEGs tested as well as increases in *Ttr* in all of the tested cell types. We have replaced the previous **Figures 7-8** with the updated images and removed IHC in **Supplementary Fig. 11** as it does not provide additional information and is not quantitative as the reviewer pointed out. We hope that our new data are now of sufficient quality.

REVIEWERS' COMMENTS:

Reviewer #1 (Remarks to the Author):

The authors adequately addressed all the concerns raised by this reviewer.

Reviewer #3 (Remarks to the Author):

The authors have satisfactorily addressed all of my concerns.

Response to Reviewer Comments

Reviewers' comments:

Reviewer #1 (Remarks to the Author): The authors adequately addressed all the concerns raised by this reviewer.

Reviewer #3 (Remarks to the Author): The authors have satisfactorily addressed all of my concerns.

Response: We are pleased that all reviewers found our revision satisfactory and we thank them for the constructive and thoughtful comments which have helped us significantly improve our study.